# Wave-triggered breakup in the marginal ice zone generates lognormal floe size distributions: a simulation study

Nicolas Mokus[1] and Fabien Montiel[1]

[1]Department of Mathematics and Statistics, University of Otago, Dunedin, New Zealand

**Correspondence:** Nicolas Mokus (nmokus@maths.otago.ac.nz)

**Abstract.** Fragmentation of the sea ice cover by ocean waves is an important mechanism impacting ice evolution. Fractured ice is more sensitive to melt, leading to a local reduction in ice concentration, facilitating wave propagation. A positive feedback loop, accelerating sea ice retreat, is then introduced. Despite recent efforts to incorporate this process and the resulting floe size distribution (FSD) into the sea ice components of global climate models (GCM), the physics governing ice breakup under wave action remains poorly understood, and its parametrisation highly simplified. We propose a two-dimensional numerical model of wave-induced sea ice breakup to estimate the FSD resulting from repeated fracture events. This model, based on linear water wave theory and visco-elastic sea ice rheology, solves for the scattering of an incoming time-harmonic wave by the ice cover and derives the corresponding strain field. Fracture occurs when the strain exceeds an empirical threshold. The geometry is then updated for the next iteration of the breakup procedure. The resulting FSD is analysed for both monochromatic and polychromatic forcings. For the latter results, FSDs obtained for discrete frequencies are combined following a prescribed wave spectrum. We find that under realistic wave forcing, lognormal FSDs emerge consistently in a large variety of model configurations. Care is taken to evaluate the statistical significance of this finding. This result contrasts with the power-law FSD behaviour often assumed by modellers. We discuss the properties of these modelled distributions, with respect to the ice rheological properties and the forcing waves. The projected output can be used to improve empirical parametrisations used to couple sea ice and ocean waves GCM components.

## 1 Introduction

Sea ice is a distinctive feature of both polar oceans and has a profound influence on our climate. It blankets a significant fraction of the Earth, is hard to reach, and offers particularly harsh fieldwork conditions. Consequently, numerical modelling is a valuable tool not only for forecasting ice extent evolution, but also to gain insights, at a global scale, into the physical processes shaping this evolution. Hindcast results straying away from observations (Stroeve et al., 2007) hint at not fully understood internal climate variability (Zhang et al., 2018; Castruccio et al., 2019) or missing physics, such as the effect of waves on the ice cover (Squire, 2020). Global climate models (GCM) have typically overlooked this impact, even if advances were made in recent years (Roach et al., 2018; Boutin et al., 2020). Waves can break the ice, especially as thinner ice becomes prevalent. For instance, the 2012 record-low Arctic sea ice extent was amplified by wave activity (Parkinson and Comiso,

2013). With thinner and weaker first-year ice becoming dominant in the Arctic (Kwok et al., 2009; Kwok, 2018), the sea ice grows more vulnerable to flexure-induced failure.

The marginal ice zone (hereafter MIZ), a belt of loosely to densely packed ice floes, serves as a buffer between the ice-free open ocean and the pack ice; it is a region notably affected by waves (Dumont et al., 2011). Tracking these individual, floating pieces of ice and characterising their dynamics at the basin scale is not possible. Since the pioneering work of Rothrock and

Thorndike (1984), researchers have taken interest in describing the floe size distribution (hereafter FSD) and its effect on the climate system. Of particular interest here, fragmentation caused by ocean waves makes the floes more sensitive to melt (Steele, 1992; Perovich and Jones, 2014), even for larger ones (Horvat et al., 2016), locally decreasing the ice concentration and allowing waves to propagate further into the MIZ. It leads to more fragmentation, thus introducing an ice–wave feedback loop (Asplin et al., 2012; Thomson and Rogers, 2014).

Remote sensing observations of floe sizes (e.g. Rothrock and Thorndike, 1984; Toyota et al., 2006; Steer et al., 2008; Toyota et al., 2011; Wang et al., 2016) have led to the rooted conception that the FSD follows a power law; an – often truncated – Pareto distribution. However, it is unclear how much waves contribute to the emergence of this power law, as wave conditions prior or during observations, as well as ice properties, are not always reported (Herman et al., 2021). A variety of other processes (such as failure from wind or internal stress, lateral melting or growth, ridging, rafting or welding) are thought to be likely to

alter the FSD (Rothrock and Thorndike, 1984). Additionally, a broad spectrum of acquisition techniques, areas and times of studies, and a priori assumptions (e.g. the parametric form of the distribution tail) led to parametrisations of this power law covering a large span of exponents. Stern et al. (2018) exposed that the widespread distribution fitting technique used, least squares regressions in log-log space, is likely to have led to significant bias in these exponent estimates.

For the last decades, wave-ice interaction has been an active field of research and numerous theoretical models, gradually

benefiting from advances in computing power, have been developed. Classically, the goal was to understand the attenuation imposed on the waves by the ice cover (Wadhams, 1973). A component of this attenuation is scattering, that results from inhomogeneities of the ice cover such as cracks and leads (Williams and Porter, 2009), changes of thickness as caused by pressure ridges (Vaughan and Squire, 2007), and floes themselves (Meylan and Squire, 1994; Kohout and Meylan, 2008; Bennetts and Squire, 2011; Montiel et al., 2016). The thin plate model, that represents floes as floating elastic plates, has been

commonly used to retrieve an attenuation rate, which has a functional dependence on frequency (e.g. Kohout and Meylan, 2008). Dissipative terms can be included in these formulations (Squire and Fox, 1992; Fox and Squire, 1994). Alternatively, rheological models seek to represent the ice cover as a viscous continuum whose properties are representative of a discontinuous field (Keller, 1998; Wang and Shen, 2010; Santi and Olla, 2017). They can be adapted to account for grease ice conditions, rather than for discrete floes, and can also include pancake ice, but it is unclear whether the benefits they bring outshine the

introduced complexity (Mosig et al., 2015). Concurrently, an extensive body of observational research (e.g. Squire and Moore, 1980; Wadhams et al., 1988; Meylan et al., 2014; Montiel et al., 2018, 2022) has been conducted and used to study attenuation.

The reciprocal response of the ice to the waves is unsatisfactorily understood, as direct observations of wave-induced floe breakup are scarce, and localised both in time and space. Numerical assessment of the feedbacks between breakup and wave propagation were pioneered by Dumont et al. (2011) and built upon by Williams et al. (2013). Various models have imple-

mented a breakup parametrisation, either to investigate the FSD (Montiel and Squire, 2017; Herman, 2017) or to evaluate the impact of its introduction on other quantities such as ice thickness or concentration (Roach et al., 2018; Bateson et al., 2020; Boutin et al., 2021). These parametrisations are usually based on a fracture criterion involving either stress (Williams et al., 2017; Montiel and Squire, 2017) or strain (Kohout and Meylan, 2008; Williams et al., 2013; Horvat and Tziperman, 2015; Boutin et al., 2018), or a combination of both (Dumont et al., 2011). When these quantities exceed a critical value, breakup is triggered. These models cover a large span of complexities, from ad hoc configuration with simplified geometry to inclusion in a global sea ice model run in stand-alone mode (Bennetts et al., 2017; Williams et al., 2017; Roach et al., 2018; Bateson et al., 2020) or coupled to other GCM components (Roach et al., 2019; Boutin et al., 2021). The inclusion of the FSD in sea ice models is therefore actively being developed.

Zhang et al. (2015) proposed a FSD theory treating breakup as a stochastic process redistributing floe sizes and abstracting the wave forcing into a model parameter depending on the local wind, ice concentration, floe size and ice thickness. The model was calibrated to reproduce observations, assuming a power law distribution of floe sizes. Boutin et al. (2018) generalised the formulations developed by Dumont et al. (2011) and Williams et al. (2013) to adapt them to the spectral wave model WAVEWATCH III® (WW3). Wave attenuation is determined by a mean floe size, computed assuming a power law FSD. Their work was extended by coupling WW3 to the sea ice models LIM3 (Boutin et al., 2020) and neXtSIM (Boutin et al., 2021). The former assumed a power-law FSD with fixed exponent, while the latter let the exponent vary based on ice conditions. Bateson et al. (2020) represent the FSD as a power law in CICE, and allow the exponent to vary seasonally or under the influence of the ice concentration. Therefore, it appears that most representations of wave-induced sea ice breakup in large-scale wave or sea ice models are based on the assumption that the FSD follows a power law. The prognostic model of Roach et al. (2018), based on the FSD modelling framework proposed by Horvat and Tziperman (2015), does not make this assumption, at the expense of computational cost. Horvat and Roach (2022) were able to reduce the additional cost attributable to the FSD evolution by training a neural network with the results from the original parametrisation.

Recent numerical experiments have been conducted to investigate the wave effect on the FSD without a priori assumptions on the distribution shape. Montiel and Squire (2017) extended the 3D linear wave scattering model of wave attenuation in the MIZ proposed by Montiel et al. (2016) by including a stress-based failure criterion. They investigated the FSD obtained after repeated breakup events and found that near normal or bimodal distributions emerged for a wide range of wave and ice conditions. However, computational constraints limited their ability to perform simulations on sufficiently large scales to conduct robust statistical analyses of these distributions. Herman (2017) coupled a non-hydrostatic, non-linear wave model to sea ice represented as bonded grains, hence relaxing assumptions inherent to potential flow theory and allowing for the computation of a transient solution; at the expense of computational efficiency, limiting the usability of the model to smaller scale configurations. The resulting FSDs were narrow, bounded distributions governed by the grain sizes. Both approaches rely on some binning of the floe sizes, either directly (Montiel and Squire, 2017) or indirectly though the use of discrete elements (Herman, 2017), effectively ensuing discrete distributions. Recent field observations of floes directly impacted by waves (Dumas-Lefebvre and Dumont, 2021; Herman et al., 2021) and laboratory experiments (Herman et al., 2018; Dolatshah et al., 2018; Passerotti et al., 2021) also suggest contrasting distributions.

In this study, we model the wave-induced breakup process with the aim of quantifying the resulting FSD. Breakup happens on time scales shorter than other processes affecting the FSD, such as thermodynamics (Collins et al., 2015), allowing us to neglect these in our model. We use the thin plate formalism in a two-dimensional setting (one horizontal, one vertical), relying on an established scattering theory, augmented with an energy dissipating process. We then incorporate a strain-based breakup parametrisation. We let an FSD emerge by repeatedly breaking off floes from a semi-infinite ice cover, and we link

the resulting distribution to the ice properties and the wave forcing. This approach is similar to the work presented by Montiel and Squire (2017); however, we simplify the geometry, stripping out one horizontal dimension, hence reducing the numerical costs to allow for the generation of more fragments. We observe that under a realistic wave forcing, our model generates FSDs appropriately described by lognormal distributions; this holds in a large span of model configurations. We discuss the effects of the wave and the ice properties on the distribution parameters. Even though we acknowledge that any parametric distribution

is likely to be an inaccurate depiction of a real ice cover, they have the advantage of efficiently encoding the information to be exchanged between GCM components (Horvat and Tziperman, 2015).

This paper unwinds as follows. In Sect. 2, we introduce notations and the underlying mathematical formulations. In Sect. 3, we describe the main components of our model, scattering and breakup. In Sect. 4 we present direct results from monochromatic simulations, upon which we build in Sect. 5 to suggest a polychromatic parametrisation. We discuss these results in Sect. 6.

**2   Preliminaries**

We consider surface gravity waves propagating in a two-dimensional fluid domain of constant, finite depth $H$ associated with a Cartesian coordinate system $(x, z)$, where $x$ and $z$ are the horizontal and vertical coordinates, respectively. Translational invariance is assumed in the second horizontal direction. We assume the fluid to be inviscid and incompressible with density $\rho_w$. The flow is assumed to be irrotational so that the fluid velocity can be described by the gradient of a scalar potential $\Phi$,

which satisfies Laplace's equation:

$$\nabla^2 \Phi = 0. \tag{1}$$

We place an array of $N_f + 1$ non-overlapping ice floes, modelled as floating visco-elastic plates, in the domain; two adjacent floes are separated by open-water. Their mechanical behaviour is determined by their density $\rho$, thickness $h$, flexural rigidity $D = \frac{Yh^3}{12(1-\nu^2)}$ (where $Y$ and $\nu$ are respectively Young's modulus and Poisson's ratio), and viscosity $\gamma$; their draught

is $d = \frac{\rho}{\rho_w}h$. Floe $j$, where $j \in \{0, \dots, N_f\}$, is located in space by the horizontal coordinate of its left edge $x_j$ (ordered so that $x_j < x_{j+1}$) and its length $L_j$, with $L_{N_f}$ being infinite. At rest, the fluid region covered by floe $j$ is encompassed in the sub-domain $\Omega_j^f = [x_j, x_j + L_j] \times [-H, -d]$. The interface with the ice $\partial\Omega_j^f$ is on $z = -d$. We denote $\Omega^f = \bigcup_0^{N_f} \Omega_j^f$ and $\partial\Omega^f = \bigcup_0^{N_f} \partial\Omega_j^f$. The ice-free sub-domain left of the floe-covered sub-domain $\Omega_j^f$ is $\Omega_j^w$ so that at rest,

$$\Omega_j^w = \begin{cases} [x_{j-1} + L_{j-1}, x_j] \times [-H, 0], & j > 0 \\ (-\infty, x_0] \times [-H, 0], & j = 0 \end{cases} . \tag{2}$$

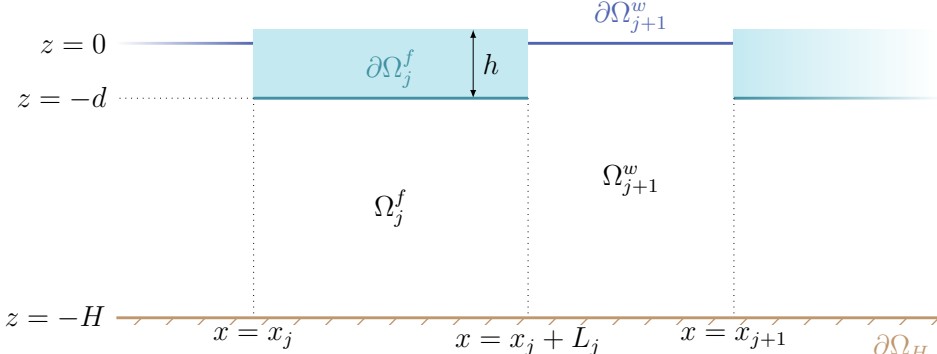

**Figure 1.** Geometry of the model at rest. Wave forcing would alter the fluid boundaries around $z = 0$ and $z = -d$.

The horizontal bottom boundary $\partial\Omega_H$ is at $z = -H$ and the interface $\partial\Omega_j^w$ between the atmosphere and $\Omega_j^w$ is at $z = 0$ when the fluid is at rest. We define $\Omega^w$ and $\partial\Omega^w$ in the same way as $\Omega^f$ and $\partial\Omega^f$. The whole fluid domain is $\Omega = \Omega^w \cup \Omega^f$. Our notations and the geometry of the model are summarized on Fig. 1.

The system is forced by a monochromatic plane wave of angular frequency $\omega$ propagating in the positive $x$ direction. The wave amplitude $a$ is assumed to be small compared to the wavelength. The perturbed top boundary of the fluid is located at

$z = \eta(x,t)$, whether it is an interface with the atmosphere (in which case $\eta \approx 0$) or with an ice floe (in which case $\eta \approx -d$). We further set $\Phi = \mathrm{Re}\left[\phi(x,z)\exp(-i\omega t)\right]$ with $\phi$ a time-independent, complex-valued function.

We consider the seabed to be impervious, hence not allowing for normal flow, so that on $\partial\Omega_H$

$$\frac{\partial\phi}{\partial z} = 0. \tag{3}$$

The small amplitude forcing allows us to use linear surface waves theory in $\Omega^w$, leading to the boundary condition

$$\frac{\partial\phi}{\partial z} = \frac{\omega^2}{g}\phi \tag{4}$$

on $\partial\Omega^w$, where $g$ is the acceleration due to gravity. We model the flexural motion of the ice floes using the modified Kirchhoff-Love plate theory, introduced by Robinson and Palmer (1990). Coupling to the fluid motion in $\Omega^f$ yields

$$\left(\frac{D}{\rho_w g}\frac{\partial^4}{\partial x^4} + 1 - \frac{\omega^2 d}{g} - i\frac{\gamma\omega}{\rho_w g}\right)\frac{\partial\phi}{\partial z} = \frac{\omega^2}{g}\phi \tag{5}$$

on $\partial\Omega^f$. Eqs. (4) and (5) stem from the assumptions that under small-amplitude wave forcing, waves do not break and fluid

is at all time in contact with the bottom of the ice. Details on the derivations can be found in e.g. Fox and Squire (1994) or Williams et al. (2013).

We also neglect the surge motion of the floes, meaning that

$$\frac{\partial\phi}{\partial x} = 0 \tag{6}$$

on $\{x_j, x_j + L_j\} \times [-d, 0]$.

We finally add the free edge conditions

$$\frac{\partial^3 \phi}{\partial x^2 z} = 0, \quad \frac{\partial^4 \phi}{\partial x^3 z} = 0 \tag{7}$$

on $\{x_j, x_j + L_j\} \times \{-d\}$, which assume that bending moment and vertical stress vanish at the floe boundaries, respectively.

The boundary conditions given in Eqs. (3)-(7) together with Eq. (1) in its time-independent form, $\nabla^2 \phi = 0$, complete our boundary value problem.

## 3   Methods

The solution to the boundary-value problem described in Sect. 2 is based on eigenfunction expansions and multiple wave scattering theory. A closely related problem (no viscous dissipation and zero-draught approximation) was considered by Kohout and Meylan (2008), which forms the basis of the wave attenuation formulation in Horvat and Tziperman (2015), and was revisited by Montiel et al. (2012) with non-zero draught. Although similar in some aspects, our method differs from these two studies in the ways scattering by a single floe edge is treated and multiple scattering is resolved.

### 3.1   Wave scattering

In any sub-domain $\Omega_j^w$ or $\Omega_j^f$, the velocity potential is decomposed as the superposition of forward-travelling and backward-travelling plane waves, using the ansatz $\left[ c^+ e^{i(kx - \theta^+)} + c^- e^{-i(kx - \theta^-)} \right] \zeta(z)$, where $c^+, c^- \in \mathbb{C}$ are coefficients to be determined and $\theta^+, \theta^- \in [0, 2\pi)$ are phase shifts introduced to simplify analytical derivations and improve numerical stability. Solving the boundary value problem described in Sect. 2 in all sub-domains, the potential is expanded into series of travelling and evanescent wave modes

$$\phi = \begin{cases} \displaystyle\sum_{n=0}^{\infty} \left[ c_{j-1,n}^{w+} e^{i(k_n^w x - \theta_{j-1,n}^{w+})} + c_{j,n}^{w-} e^{-i(k_n^w x - \theta_{j,n}^{w-})} \right] \zeta_n^w(z), & (x, z) \in \Omega_j^w \\ \displaystyle\sum_{n=-2}^{\infty} \left[ c_{j,n}^{f+} e^{i(k_n^f x - \theta_{j,n}^{f+})} + c_{j,n}^{f-} e^{-i(k_n^f x - \theta_{j,n}^{f-})} \right] \zeta_n^f(z), & (x, z) \in \Omega_j^f \end{cases} \tag{8}$$

where superscripts $w$ or $f$ are related to an open-water or floe-covered sub-domain.

The wave numbers $\{k_n^w \mid n \geq 0\}$ are the roots of the dispersion relation

$$k \tanh(kH) = \frac{\omega^2}{g} \tag{9}$$

such that $k_0^w$ is a positive real number (therefore associated with left- and right-propagating wave modes in $\Omega^w$) while $\{k_n^w \mid n > 0\}$ are purely imaginary numbers with positive imaginary part, sorted by ascending imaginary part (associated with exponentially decaying evanescent wave modes).

Likewise, the wave numbers $\{k_n^f \mid n \geq -2\}$ are the roots of the dispersion relation

$$\left( \frac{D}{\rho_w g} k^4 + 1 - \frac{\omega^2 d}{g} - i \frac{\gamma \omega}{\rho_w g} \right) k \tanh(k(H - d)) = \frac{\omega^2}{g} \tag{10}$$

such that $\{k_n^f \mid n > -2\}$ are complex numbers in the first quadrant of the complex plane, sorted by ascending imaginary part for $n > 0$, and $k_{-2}^f$ is a complex number in the second quadrant of the complex plane.

Since $\mathcal{O}\left[\mathrm{Re}\left(k_0^f\right)\right] \gg \mathcal{O}\left[\mathrm{Im}\left(k_0^f\right)\right]$, $k_0^f$ is associated with left- and right-propagating wave modes in $\Omega^f$. On the contrary, $\mathcal{O}\left[\mathrm{Re}\left(k_n^f\right)\right] \ll \mathcal{O}\left[\mathrm{Im}\left(k_n^f\right)\right]$ for $n > 0$: these modes are associated with exponentially decaying wave modes. Finally, $\mathcal{O}\left[\mathrm{Re}\left(k_n^f\right)\right] = \mathcal{O}\left[\mathrm{Im}\left(k_n^f\right)\right]$ for $n \in \{-2, -1\}$, so these two roots are associated with attenuating, propagating wave modes. Details on these behaviours can be found in Williams et al. (2013). In the special case where $\gamma = 0$ (purely elastic floes) then $k_0^f$ is a positive real number, $k_{-1}^f$ is in the first quadrant of the complex plane, $k_{-2}^f = -\overline{k_{-1}^f}$, and $\{k_n^f \mid n > 0\}$ are purely imaginary numbers with positive imaginary part. We note that when $1 - \frac{\omega^2 d}{g}$ becomes negative (large frequencies or thickness), instead of having the three distinct roots $k_{-2}^f$, $k_{-1}^f$ and $k_0^f$, Eq. (10) may admit one double root or one triple root (Williams, 2006, p. 39) as the complex roots may become purely imaginary. Therefore, we enforce $\omega \leq \sqrt{\frac{g}{d}}$ in this study to avoid this issue. For an ice thickness of $1\,\mathrm{m}$, it corresponds to a minimum admissible period of approximately $1.90\,\mathrm{s}$.

For any wave mode $n$, floe $j$ radiates four waves: two from its left edge (with coefficients $c_{j,n}^{w-}$, $c_{j,n}^{f+}$) and two from its right edge (with coefficients $c_{j,n}^{w+}$, $c_{j,n}^{f-}$) except for the leftmost, semi-infinite floe ($j = N_f$) whose right edge is ignored. The coefficients $c_{-1,n}^{w+}$ and $c_{N_f,n}^{f-}$ are prescribed and represent the right-travelling incident wave forcing in $\Omega_0^w$ and the absence of forcing in $\Omega_{N_f}^f$, respectively: only $c_{-1,0}^{w+} = -i\frac{g}{\omega}a$ is non-zero. The quantity $\theta = \theta_{-1,0}^{w+}$ is an arbitrary phase associated with the forcing. We also note that $\{\theta_{-1,n}^{w+} \mid n > 0\}$ and $\{\theta_{N_f,n}^{f-} \mid n \geq -2\}$ do not take part in the computation and are left undefined. The remaining phases are determined so that the exponential terms in Eq. (8) reduce to 1 when evaluating $\phi$ at the edge radiating the wave, i.e.

$$\theta_{j,n}^{w+} = k_n^w\left(x_j + L_j\right) \quad ; \quad \theta_{j,n}^{w-} = k_n^w x_j \quad ; \quad \theta_{j,n}^{f+} = k_n^f x_j \quad ; \quad \theta_{j,n}^{f-} = k_n^f\left(x_j + L_j\right). \tag{11}$$

Finally, the functions

$$\zeta_n^w(z) = \frac{\cosh(k_n^w(H+z))}{\cosh(k_n^w H)}, \ z \in [-H, 0] \quad ; \quad \zeta_n^f(z) = \frac{\cosh(k_n^f(H+z))}{\cosh(k_n^f(H-d))}, \ z \in [-H, -d] \tag{12}$$

are vertical basis functions in the free-surface sub-domains and the ice-covered sub-domains, respectively.

### 3.1.1 Scattering by one floe edge

We obtain the solution to the multiple scattering problem of the incident wave by the ice floes by imposing continuity of pressure and normal velocity across the vertical boundaries between adjacent ice-free and ice-covered sub-domains, i.e. at each floe edge. These conditions are enforced by matching $\phi$ and $u = \frac{\partial \phi}{\partial x}$ on both sides of each interface. The single edge matching problem is solved using an integral equation method, as described by Williams and Porter (2009) and Mosig (2018).

Considering the scattering by the left edge of floe $j$ and assuming knowledge of $c_{m-1,n}^{w+}$ and $c_{m,n}^{f-}$, the method generates a set of scattering relations relating these incident wave modes coefficients to those associated with wave modes propagating or decaying away from the edge, $c_{j,n}^{f+}$ and $c_{j,n}^{w-}$. When truncating the series in Eq. (8) to $N_v$ evanescent modes, the relations can

be summarised by the matrix equation

$$\begin{pmatrix} \boldsymbol{c}_j^{f+} \\ \boldsymbol{c}_j^{w-} \end{pmatrix} = \begin{pmatrix} \mathbf{T}_j^{fw} & \mathbf{R}_j^f \\ \mathbf{R}_j^w & \mathbf{T}_j^{wf} \end{pmatrix} \begin{pmatrix} \mathbf{S}_j^w & \mathbf{0} \\ \mathbf{0} & \mathbf{S}_j^f \end{pmatrix} \begin{pmatrix} \boldsymbol{c}_{j-1}^{w+} \\ \boldsymbol{c}_j^{f-} \end{pmatrix}, \tag{13}$$

where $\mathbf{T}^{fw} \in \mathbb{C}_{N_v+3 \times N_v+1}$, $\mathbf{T}^{wf} \in \mathbb{C}_{N_v+1 \times N_v+3}$, $\mathbf{R}^f \in \mathbb{C}_{N_v+3 \times N_v+3}$, $\mathbf{R}^w \in \mathbb{C}_{N_v+3 \times N_v+3}$ are matrices respectively describing transmission and reflection of waves in either directions through the floe edge, $\boldsymbol{c}_j^{w\pm} = \left( c_{j,0}^{w\pm}, \ldots, c_{j,N_v}^{w\pm} \right)^{\mathrm{T}}$, $\boldsymbol{c}_j^{f\pm} = \left( c_{j,-2}^{f\pm}, \ldots, c_{j,N_v}^{f\pm} \right)^{\mathrm{T}}$ are vectors of unknown coefficients, and $\mathbf{S}^w \in \mathbb{C}_{N_v+1 \times N_v+1}$, $\mathbf{S}^f \in \mathbb{C}_{N_v+3 \times N_v+3}$ are diagonal phase shift matrices.

By symmetry, the scattering by the right edge of floe $j$ is described by

$$\begin{pmatrix} \boldsymbol{c}_j^{w+} \\ \boldsymbol{c}_j^{f-} \end{pmatrix} = \begin{pmatrix} \mathbf{T}_j^{wf} & \mathbf{R}_j^w \\ \mathbf{R}_j^f & \mathbf{T}_j^{fw} \end{pmatrix} \begin{pmatrix} \mathbf{S}_j^f & \mathbf{0} \\ \mathbf{0} & \mathbf{S}_{j+1}^w \end{pmatrix} \begin{pmatrix} \boldsymbol{c}_j^{f+} \\ \boldsymbol{c}_{j+1}^{w-} \end{pmatrix}. \tag{14}$$

The reflection and transmission matrices depend only on the quantities present in the dispersion relations, Eqs. (9) and (10). We assess convergence of the numerical procedure by investigating energy conservation after scattering for floes of zero viscosity. Our analysis proved $N_v = 2$ to be adequate. Therefore, we use it in the rest of this study.

### 3.1.2 Scattering by an array of floes

Wave fields radiated by adjacent floes are coupled, which is clearly shown by Eqs. (13) and (14). To solve the multiple scattering problem described by these equations, we take advantage of the sparsity of the matrix representing the combined linear system, using a dedicated solver (Demmel et al., 1999; Virtanen et al., 2020). Specifically, we solve

$$\mathbf{M}\boldsymbol{c} = \boldsymbol{f} \tag{15}$$

where $\mathbf{M}$ is a tridiagonal block matrix and $\boldsymbol{c}$ the vector of unknown potential coefficients.

An array of $N_f + 1$ *finite* floes leads to the matrix

$$\tilde{\mathbf{M}} = \begin{pmatrix} \mathbf{M}_0 & \mathbf{U}_1 & \mathbf{0} & \cdots & & \mathbf{0} \\ \mathbf{L}_1 & \mathbf{M}_1 & \mathbf{U}_2 & & & \vdots \\ \mathbf{0} & \mathbf{L}_2 & \mathbf{M}_2 & \ddots & & \mathbf{0} \\ \vdots & & \ddots & \ddots & & \mathbf{U}_{N_f} \\ \mathbf{0} & \cdots & & \mathbf{0} & \mathbf{L}_{N_f} & \mathbf{M}_{N_f} \end{pmatrix} \tag{16}$$

with each block element of $\tilde{\mathbf{M}}$ is a square matrix of size $4(N_v + 2)$; $\mathbf{0}, \mathbf{1}$ denote 0-filled matrices and identity matrices, respectively, with sizes compatible with other matrices in the same rows and columns.

As the last floe is here *infinite*, we obtain $\mathbf{M}$ introduced in Eq. (15) by trimming down $\tilde{\mathbf{M}}$ from its last $2(N_v + 2)$ rows and columns. While the size of $\mathbf{M}$ is $[2(N_v + 2)(2N_f + 1)]^2$, it has only $4(N_v + 2)\left[N_f(2N_v + 5) + \frac{1}{2}\right]$ non-zero elements: this number grows linearly with $N_f$, instead of quadratically.

The vector of unknown coefficients is

$$\boldsymbol{c} = \begin{pmatrix} \boldsymbol{c}_0 & \cdots & \boldsymbol{c}_{N_f-1} & \boldsymbol{c}_{N_f}^{w-} & \boldsymbol{c}_{N_f}^{f+} \end{pmatrix}^T, \quad \text{with } \boldsymbol{c}_j = \begin{pmatrix} \boldsymbol{c}_j^{w-} & \boldsymbol{c}_j^{f+} & \boldsymbol{c}_j^{f-} & \boldsymbol{c}_j^{w+} \end{pmatrix}^T, \tag{17}$$

and the forcing term

$$\boldsymbol{f} = e^{i\theta} \begin{pmatrix} \mathbf{R}_0^w \boldsymbol{c}_{-1}^{w+} & \mathbf{T}_0^{wf} \boldsymbol{c}_{-1}^{w+} & \mathbf{0} & \cdots & \mathbf{0} \end{pmatrix}^T. \tag{18}$$

Building $\mathbf{M}$ and $\boldsymbol{f}$ and solving for $\boldsymbol{c}$ is linear in time for $\mathcal{O}(N_f) > 10$. For $N_f = 10^5$, these operations are done in around
230 $10\,\mathrm{ms}$ on an Intel Core i5-6300U 2016 laptop. Solving Eq. (15) for $\boldsymbol{c}$ fully determines the spatial part of the potential field in $\Omega$.

## 3.2 Breakup parametrisation

To parametrise the breakup, we build upon the commonly used strain-based approach (e.g., Kohout and Meylan, 2008; Horvat and Tziperman, 2015). As opposed to these authors' approaches, we account for each floe individually rather than parametrising
the strain decay from the number of floes or considering the ice cover to be continuous.

When using the plane stress approximation and our symmetry assumption, the strain $\tilde{\varepsilon}_j$ undergone by ice floe $j$ is

$$\tilde{\varepsilon}_j(x', z') = -z' \frac{\partial^2 w}{\partial x'^2} \tag{19}$$

where $(x', z')$ describes a coordinate system local to the floe, defined as $x' = x - x_j$ and $z' = z - \left(\frac{h}{2} - d\right) = z - h\left(\frac{1}{2} - \frac{\rho}{\rho_0}\right)$, hence setting the origin on the intersection of the floe's left edge and horizontal middle surface. Under the plane stress approx-
240 imation, the vertical displacement field undergone by the floe, $w(x,t)$, does not depend on $z'$: $w(x,t) = \eta(x,t)$. As $|z'| \leq \frac{h}{2}$, the maximum (in absolute value) strain, $\varepsilon_j$, is located on either surface of the floe, i.e. $z' = \pm\frac{h}{2}$. It follows that

$$\varepsilon_j(x') = \frac{h}{2}\left|\frac{\partial^2 \eta}{\partial x'^2}\right| = \frac{hT}{4\pi}\left|\mathrm{Re}\left[i\frac{\partial^2}{\partial x'^2}\frac{\partial \phi}{\partial z}\right]\right| \tag{20}$$

with $T = \frac{2\pi}{\omega}$ the wave period.

Floe $j$ is set to break when

$$\max_{x' \in [0, L_j]} \varepsilon_j(x') > \varepsilon_c \tag{21}$$

where $\varepsilon_c$ is an empirically determined strain threshold.

We situate the breakup point at

$$x_b = \mathrm{argmax}\,\varepsilon_j \tag{22}$$

so that a floe of length $L_j$ is turned into two floes of length $x_b$ and $L_j - x_b$. Hence, the number of floes at most doubles, if all
250 the floes break in a single breakup simulation.

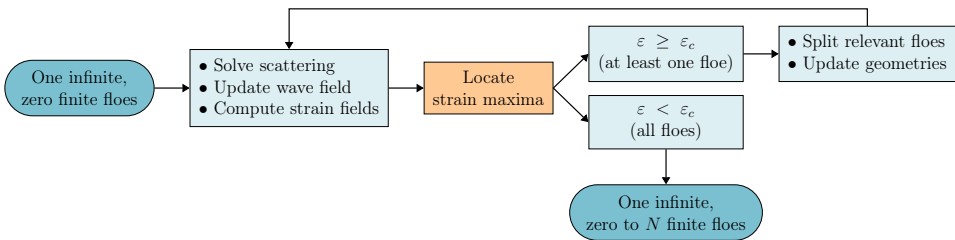

**Figure 2.** Outline of the numerical experiment.

### 3.3 Numerical experiment set-up

The values of the parameters kept fixed across all simulations are given in Table 1. Our experiments unwind as follows:

1. The model is initialised with a set of physical parameters as input and a single, semi-infinite floe.

2. The scattered wave field is determined, and the strain field evaluated for every floe in the domain. All the floes for which the conditions for breakup are met are split, and the domain is updated.

3. The second step is repeated until none of the floes break or a prescribed number of iterations reached.

These steps are summarized in Fig.2. The output is the set of coordinate-length pairs describing the final geometry.

For a given iteration, all floes are scrutinised for breakup first, then their positions are updated. As we neglect floe motion, to prevent them from overlapping, they are assigned order-preserving random locations, determined through a process described in Appendix A. As we do not build any energy dissipation mechanism for the fluid in $\Omega^w$, the width of the gap between the floes impacts the phase of the wave, but not its amplitude as it reaches floe $j+1$ after transmission by floe $j$. Thus, whether floe $j+1$ breaks or not is unlikely (insomuch as its length allows sufficient strain to be reached) to be affected by this gap, but the breakup location may be. To account for this introduced randomness, we run each simulation as an ensemble with 50 separate realisations. As a side benefit, doing so removes the possible effect of local resonances in any single realisation, as described by Kohout and Meylan (2008). We ran a sensitivity analysis (see Appendix A) to confirm that the choice of the parameters governing the random placement does not affect the resulting distribution of floe lengths.

We note that in the arrangement considered here, the transmission and reflection matrices (Eqs. (13) and (14)) are the same for every floe: they need only be computed once. This stems from the experiment design, as all floes are assumed to inherit their physical properties (i.e. thickness, density, rigidity and viscosity) from the parent semi-infinite floe used to initialise the simulation. These properties could be altered after the breakup, but we choose to keep them constant as breakup happens on time scales shorter than those of the processes that would e.g. alter the thickness. Our model can however handle a general case with floes of varying thickness, flexural rigidities, densities and viscosities. Such simulations are outside the scope of this paper. The information carried by the positions and dimensions of the floes are encapsulated in the phase shift matrices (Eqs. (13) and (14)) which are truly floe-dependent.

**Table 1.** Fixed model parameters and their values.

| Symbol | Name | Value |
|--------|------|-------|
| $g$ | Acceleration of gravity | $9.8\,\mathrm{m\,s^{-2}}$ |
| $\rho$ | Ice density | $922.5\,\mathrm{kg\,m^{-3}}$ |
| $\gamma$ | Ice viscosity | $20\,\mathrm{Pa\,s\,m^{-1}}$ |
| $\nu$ | Ice Poisson's ratio | $0.3$ |
| $Y$ | Ice Young's modulus | $6\,\mathrm{GPa}$ |
| $\rho_w$ | Ocean density | $1025\,\mathrm{kg\,m^{-3}}$ |
| $H$ | Ocean depth | $2400\,\mathrm{m}$ |

The final result extracted from the simulation is the set of newly formed floe lengths, excluding the semi-infinite floe on the right of the domain, considered as a steady state FSD.

## 4 Monochromatic forcing

We first investigate the FSD our model generates under monochromatic forcing with prescribed wave period $T$, corresponding to angular frequency $\omega = \frac{2\pi}{T}$. In addition to wave frequency, we seek to characterise the effect of the ice thickness $h$ and the

280 strain breakup threshold $\varepsilon_c$ on the FSD.

Figure 3 (a,b) show example histograms of FSDs obtained for $T = 8\,\mathrm{s}$ and $a = 50\,\mathrm{cm}$, while Fig. 3 (c,d) show the influence of $T$ on the spread of the FSD. Strain thresholds considered span the range of observed values reported from scarce field measurements (Kohout and Meylan, 2008).

We obtain unimodal, right-skewed histograms (Fig. 3 (a,b)), appropriately fitted by lognormal distributions (more details

on this distribution are presented in Sect. 5.2). In the following, we use the median and the interquartile ranges as measures of central tendency and spread, respectively (Fig. 3 (c,d)). We note that for right-skewed distribution, the median is bounded below by the mode of the distribution, and bounded above by its mean. In this Section, and in the following, we display so-called number distributions, such that the area of one histogram bins is the fraction of the obtained floes whose dimensions fall within the bin. Alternatively, considering so-called area distributions, where observations are weighted by their area (which

we assume proportional to the length squared), alters the shape so that the peak of the distribution is shallower and shifted towards larger floes, and the tail of the distribution thicker. The previous comments on unimodality, skewness and lognormal fits, however, still hold.

Increasing $\varepsilon_c$ has only a moderate effect on the shape of the FSD (the skewness being respectively $1.48 \pm 0.15$, $1.46 \pm 0.23$, $1.63 \pm 0.37$ for increasing values of $\varepsilon_c$ displayed in Fig. 3 (a)), a more noticeable effect being a shift towards larger floes.

However, increasing $h$ from $50\,\mathrm{cm}$ to $1\,\mathrm{m}$ has a more dramatic influence on the FSD, shifting it towards larger floes, increasing

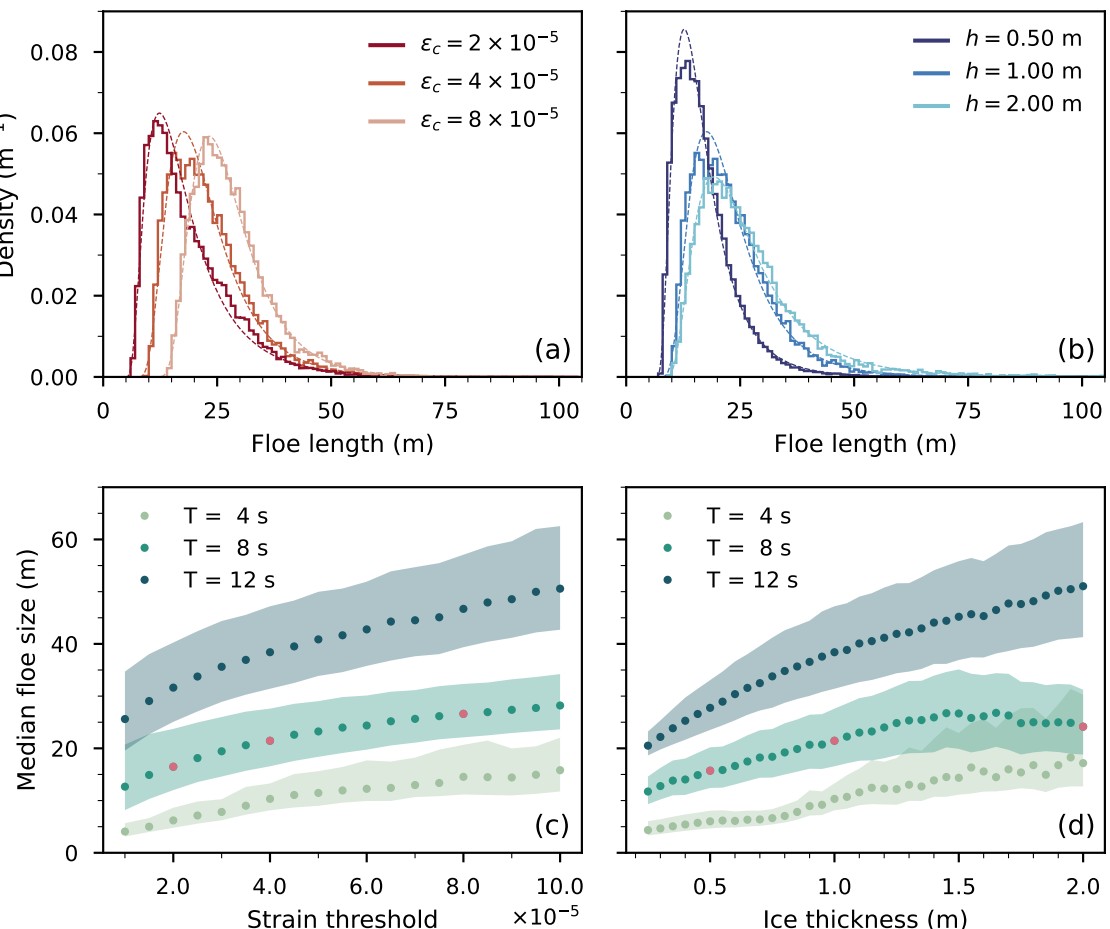

**Figure 3.** Impact of varying $T$, $\varepsilon_c$ and $h$ on the FSD with $a = 50\,\mathrm{cm}$. (a) Histograms of the floe length for different $\varepsilon_c$. The bin width is $1\,\mathrm{m}$ and $T = 8\,\mathrm{s}$. Lognormal fits are superimposed in dashed lines. (b) Same as (a) but for different $h$. (c) Evolution of the median floe size, when increasing $\varepsilon_c$, for different $T$. The shaded area indicates the corresponding interquartile range. (d) Same as (c) but for increasing $h$. In (a,c) $h = 1\,\mathrm{m}$; in (b,d) $\varepsilon_c = 4 \times 10^{-5}$. In the top panels, darker hues denote weaker ice (higher strain threshold, lower thickness) and lighter hues denote stronger ice. In the bottom panels, the contrasting dots indicate the values plotted in the corresponding top panels. All plotted quantities are ensemble averages over 100 realisations.

the spread and thickening its tail. Doubling the thickness a second time brings the same sort of changes, in a less pronounced fashion. Therefore, an increased ice mechanical resistance (either through its thickness or its strain threshold) leads to the presence of larger floes. This is expected, as the free edge boundary conditions (Eq. (7)) impose zero-strain at the floe edges, thus involving longer stretches of ice to reach higher absolute strains.

Figure 3 (c,d) shows two clear trends. Both longer waves and stronger ice lead to larger median floe sizes and wider interquartile ranges, at the exception of $T = 8\,\mathrm{s}$ in panel (c), that shows a decrease in spread with increasing $\varepsilon_c$. It shows that

wave properties alone do not govern the dominant floe size. However, the trend in median floe size is more ambiguous at high thickness. The apparent oscillations observed at $h > 1.5\,\mathrm{m}$, $T = 4\,\mathrm{s}$ are a spurious consequence of the fairly low amount of breakup observed in these configurations (Fig. 4 (b)), enhancing the apparent variability. Similarly, at $T = 8\,\mathrm{s}$, there is an order of magnitude drop in the number of floes when $h$ increases from $1.5\,\mathrm{m}$ to $2\,\mathrm{m}$, which may cause the tapering off and slight decrease observed in the median floe size. Notably, median floe sizes are much smaller than the corresponding half-wavelength (respectively 12, 50, 112 m for periods of 4, 8, 12 s), as observed by Herman et al. (2021).

The final number of floes (number of floes reached when the forcing wave field no longer breaks any floe during a simulation) depends sharply on $T$, as shown on Fig. 4 (a,b). Three regimes can be identified: a rapid increase with $T$ for lower periods (higher frequencies), then a plateau phase, preceding a sudden decrease at higher periods (lower frequencies). The precise delimitations of these regimes depend on the ice properties, and can be explained by the non-linear relationship between $T$, $h$, and the undergone strain. Increasing $h$ or $T$ explicitly increases the maximum strain (Eq. (20)). However, increasing $T$ leads to a longer wavelength, translating to a decline in magnitude of the surface curvature term, offsetting the increase. Additionally, waves propagate with a longer wavelength under the ice than in open water: the thicker the ice, the longer the wavelength becomes. Therefore, increasing $h$ also decreases the surface curvature. Lastly, the fraction of wave energy transmitted by a floe edge, close to 1 for longer waves, drops as period decreases (Fox and Squire, 1990); the precise magnitude of this dip depends on floe length. When considering multiple scattering, these reflections exponentially stack up, making a few floes an effective barrier to low-period waves propagation.

The minimum floe size, shown in Fig. 4 (c,d), also follows three regimes roughly delimited by the same boundaries: a sharp decrease when increasing $T$ from the lowest periods, then a steady growth (appearing to be evolving linearly with the wavelength) from a local minimum, corresponding to the plateau, and a more abrupt increase corresponding to the drop at higher periods. We observe this qualitative behaviour independently of the values of $\varepsilon_c$ or $h$ considered. An increase in minimum floe size with the wavelength is expected. The initial decrease, for shorter wavelength, corresponds to simulations with a very small number of floes: as aforementioned, floes effectively reflect waves of low period. Short waves are not able to fragment further the first initial pieces that broke off the semi-infinite cover, which leads to these higher minima.

The breakup width, here defined as the cumulated length of ice broken off the semi-infinite floe and shown in Fig. 4 (e,f), is clearly concave down and peaks for wavelengths corresponding to the plateau phase. Interestingly, if a lower strain threshold leads to wider MIZs, so does thicker ice. If thick ice, when broken up, spawns less floes than thin ice, and needs higher wavelength to be bent enough to break, it allows for breakup over a much larger wavelength band and produces longer floes. Counter-intuitively, the consequence is that intermediate waves are able to propagate much further into a thicker ice cover.

## 5  Polychromatic forcing

Our model, as described in Sect. 2 and Sect. 3, parametrises the wave forcing with a single amplitude-frequency pair. In this Section, we propose an approach to extend our results to a polychromatic wave forcing.

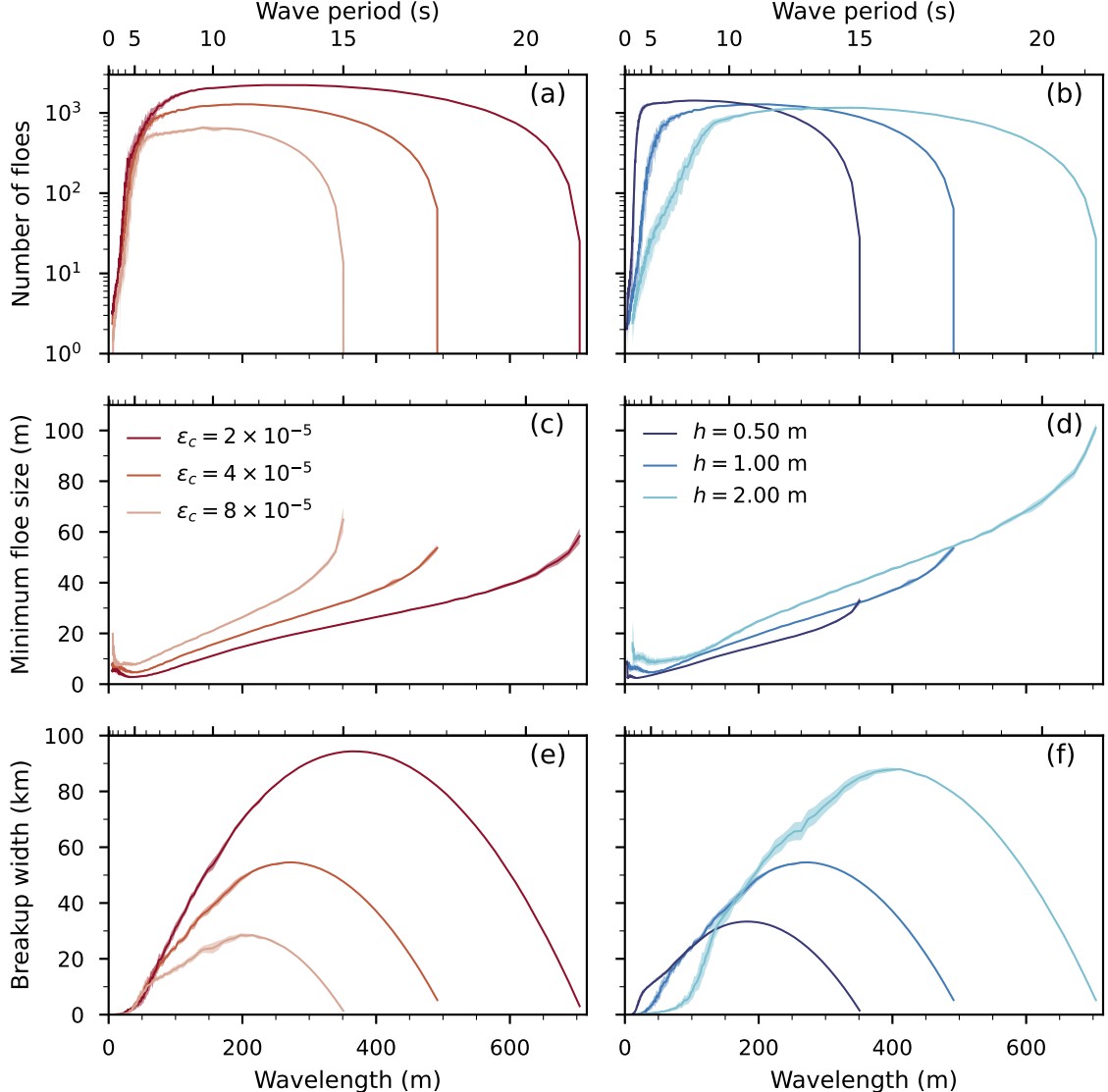

**Figure 4.** Various metrics at the end of a simulation. (a) Number of floes for different $\varepsilon_c$. (b) Same as (a) but for different $h$. (c) Minimum floe size in the sample for different $\varepsilon_c$. (d) Same as (c) but for different $h$. (e) Breakup width for different $\varepsilon_c$. (f) Same as (e) but for different $h$. The breakup width is defined as the cumulated length of the broken off floes. In (a,c,e) $h = 1\,\text{m}$; in (b,d,f) $\varepsilon_c = 4 \times 10^{-5}$; $T = 8\,\text{s}$, $a = 50\,\text{cm}$ for all panels. The wavelength axis corresponds to the open-water propagating wavelength at a given $T$. Darker hues denote weaker ice (higher strain threshold, lower thickness) and lighter hues denote stronger ice. The plotted lines are ensemble average, the shaded areas indicate one standard deviation.

## 5.1 Wave spectrum

To estimate the effect of a developed sea on the FSD $f_L(l)$, we take the weighted average of distributions $\tilde{f}_L(l,\omega)$ resulting from monochromatic model runs, with weights taken from the energy density $S(\omega)$ of a theoretical ocean spectrum over a truncated frequency interval $[\omega_{\min}, \omega_{\max}]$ so that

$$f_L(l) = \frac{\int_{\omega_{\min}}^{\omega_{\max}} \tilde{f}_L(l,\omega) S(\omega) \mathrm{d}\omega}{\int_{\omega_{\min}}^{\omega_{\max}} S(\omega) \mathrm{d}\omega}. \tag{23}$$

This FSD model assumes that different frequency components of the wave spectrum affect the FSD independently from each other. This is a strong assumption, the validity of which, will be discussed in Sect. 6. Several expressions exist for $S$; we choose to use a Pierson-Moskowitz spectrum (Pierson and Moskowitz, 1964) as it can be easily adapted to depend only on the significant wave height $H_s$ (Ochi, 2005), giving

$$S(\omega) = c_1 \frac{g^2}{\omega^5} \exp\left(-c_2 \frac{g^2}{\omega^4 H_s^2}\right) \tag{24}$$

where $c_1 = 8.1 \times 10^{-3}$, $c_2 = 3.24 \times 10^{-2}$ are non-dimensional constants. This spectrum has been used in previous wave-sea ice interaction studies (e.g., Kohout and Meylan, 2008; Dumont et al., 2011) and is a reduced version of the two-parameter Bretschneider spectrum, used in this context as well (e.g., Horvat and Tziperman, 2015, 2017; Montiel and Squire, 2017).

We evaluate Eq. (23) numerically on 200 linearly spaced frequency bins, setting $H_s = 2a$. As by definition

$$H_s = 4\sqrt{\int_0^{+\infty} S(\omega) \mathrm{d}\omega}, \tag{25}$$

the bounds of integration $\omega_{\min}$ and $\omega_{\min}$ are set so that the tails $\frac{16}{H_s^2} \int_0^{\omega_{\min}} S(\omega) \mathrm{d}\omega = \frac{16}{H_s^2} \int_{\omega_{\max}}^{+\infty} S(\omega) \mathrm{d}\omega = 5 \times 10^{-7}$, which captures a significant part of the spectrum. If necessary, $\omega_{\max}$ is adjusted to ensure $\omega \leq \sqrt{\frac{g}{d}}$, as discussed in Sect. 3.1.

For each one of the 200 frequencies, we draw an FSD $\tilde{f}_L$ at random from the 50 realisations of this configuration. These FSDs are combined as aforementioned. We repeat (with replacement) this random drawing stage 500 times in order to constitute a distribution ensemble from which statistics can be derived. Proceeding like so allows us to observe variations between different $f_L$. As we have 50 independent realisations for each of the 200 frequencies, we have virtually infinitely many ways to build $f_L$ ($50^{200} \approx 6 \times 10^{339}$).

## 5.2 Reference configuration

We consider a reference configuration where $H_s = 1\,\mathrm{m}$, $h = 1\,\mathrm{m}$, $\varepsilon_c = 4 \times 10^{-5}$; this gives us frequency bounds that correspond to the wave period varying from $1.90$ to $9.23\,\mathrm{s}$. In Sect. 5.4, we discuss variations around this scenario.

The resulting FSD, shown in Fig. 5 (a), is remarkably well fitted by a three-parameter lognormal distribution. A random variable $L$ is said to follow such a distribution with parameters $\mu, \sigma^2, \tau$ if $\log(L - \tau)$ is normally distributed with mean $\mu$ and variance $\sigma^2$:

$$L \sim \mathcal{LN}\left(\mu, \sigma^2, \tau\right) \Leftrightarrow \log(L - \tau) \sim \mathcal{N}\left(\mu, \sigma^2\right). \tag{26}$$

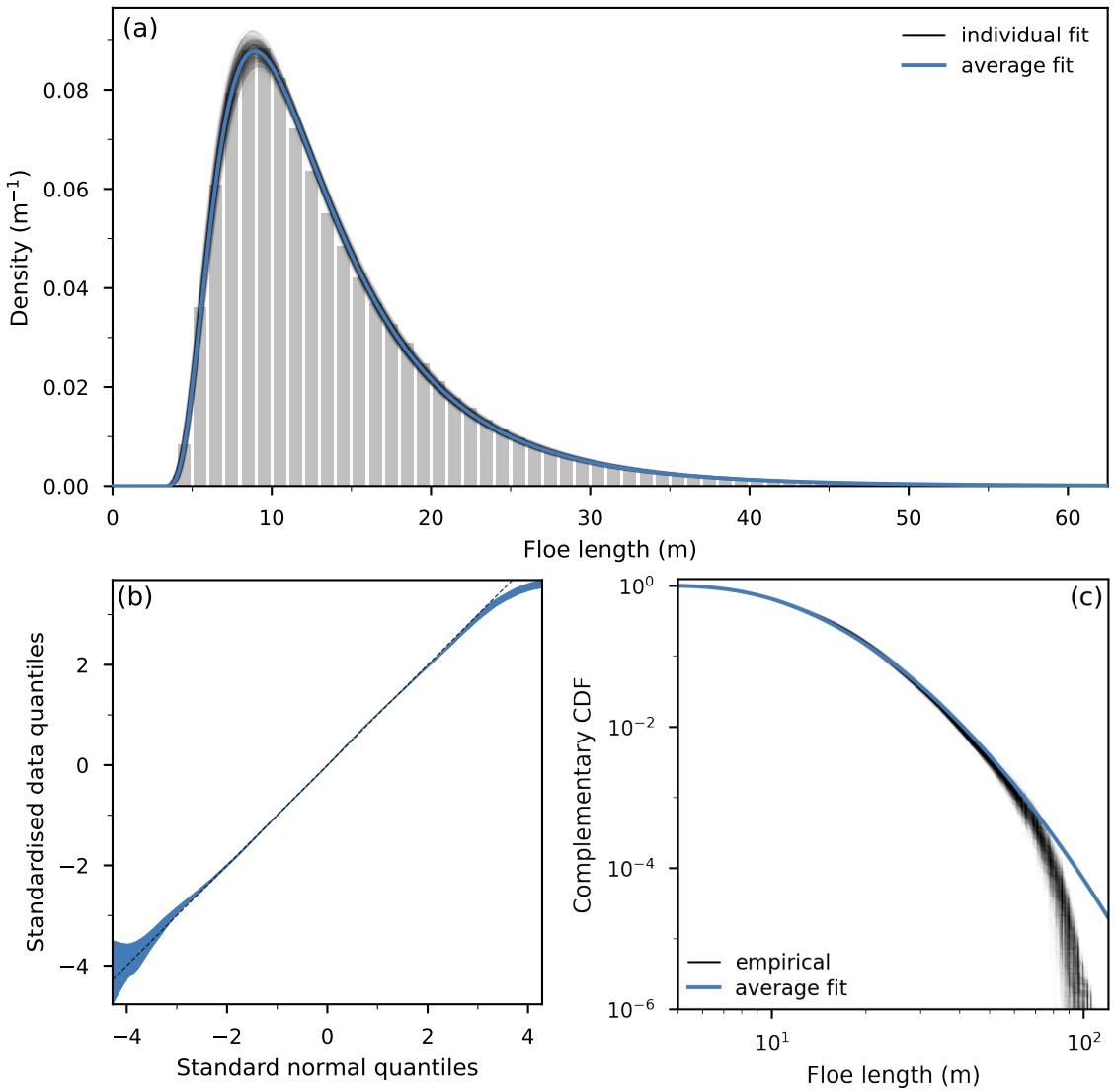

**Figure 5.** (a) FSDs for the reference configuration: lognormal fits to separate ensemble elements (thin black lines) and average distribution as detailed in text (thicker blue line). The underlying histogram depicts the ensemble average distribution. (b) Normal quantile-quantile plots of the standardised data: the coloured area corresponds to one standard deviation around the mean of the quantile-quantile lines; the dashed line indicates the main diagonal. (c) Cumulative FSDs: individual empirical CDFs (thin black lines) and CDF of the average distributions represented in (a) (thicker blue line).

The associated density function $f_L(l)$ is positive for $l > \tau$: hence $\tau$ is the smallest floe size describable by this statistical model. The scale parameter $s = \exp \mu$ has the physical dimension of the random variable – in this study, a length. The median and the mode of the distribution are given by

$$\text{median} = \tau + s \quad ; \quad \text{mode} = \tau + s \exp\left(\sigma^{-2}\right). \tag{27}$$

More details on the lognormal distribution can be found in Crow and Kunio (1988). In the following, we use the notation $\theta = (s, \sigma, \tau)$ for the parameter vector.

We obtain a point estimate $\hat{\theta}$ with maximum likelihood estimation (see e.g. Azzalini, 1996; Crow and Kunio, 1988). Among the ensemble, $\hat{\theta}$ seems to be normally distributed, with strong correlations between the three parameters and small variances. Therefore, we use the mean vector $\bar{\theta}$ to parametrise an underlying representative lognormal distribution, depicted on Fig. 5 (a,c) as thick blue lines. The linear combination of lognormal distributions does not have a simplified expression and is not, generally, a lognormal. By defining the mean distribution as a lognormal parametrised with $\bar{\theta}$, we ensure this model is preserved. This can be justified by the low spread of $\hat{\theta}$. In that regard, our random sampling aims at providing a confidence interval on the parameters values. The transformation between $\sigma^2$ and $\sigma$ and between $\mu$ and $s$ are obviously not linear. However, for the range of values considered here, we find that averaging before or after taking the transformation leads to an absolute relative difference of less than $1\%$ (median for $s$: $4 \times 10^{-2}\%$; median for $\sigma$: $4 \times 10^{-2}\%$).

We outline the goodness of fit with a quantile-quantile plot shown in Fig. 5 (b). We standardised the data using (26) before deriving the quantiles, to ease the comparison between the 500 ensemble elements and use the symmetry property of the normal distribution. Therefore, an ideal match would have the data lying on the main diagonal. We observe departure from this line for larger floes, the shallower slope suggesting the lognormal parametrisation over-predicts large floes that are not generated by the numerical model. However, as $68\%$ of the theoretical distribution belong between the $\pm 1$ ticks of the horizontal axis, and $95\%$ belong between the $\pm 2$ ticks, we deem the fit to be excellent. Figure 5 (c) is another visualisation of the goodness of fit, comparing the (strictly speaking, complementary) empirical cumulative distribution functions (CDFs) to the CDF of the theoretical average lognormal distribution. They diverge significantly only from the $10^{-2}$ tick, indicating agreement for more than $99\%$ of the range of the data. As shown in Fig. 5 (c), the CDF of a random variable following a lognormal distribution could easily be misinterpreted as piecewise straight lines when represented on a log-log plot. This kind of graph is often used for field observations, and is at the root of the ingrained power law or split power law conclusion.

## 5.3 Sub-domain FSD evolution

We do not expect the FSD to have the same shape all across an ice pack or even across the MIZ. To illustrate this effect, we analyse the evolution of the distribution when considering subsets of the domain.

Our experiment design generates distributions in parallel, for various periods in the spectrum. Therefore, there is no unique definition of the breakup width. To circumvent this issue, we use a sliding window whose bounds are relative to the local breakup width for each period used in the discretisation of the spectrum. We estimate the density $\tilde{f}_{L; b_{\text{inf}} - b_{\text{sup}}}$, with $0 \leq b_{\text{inf}} <$

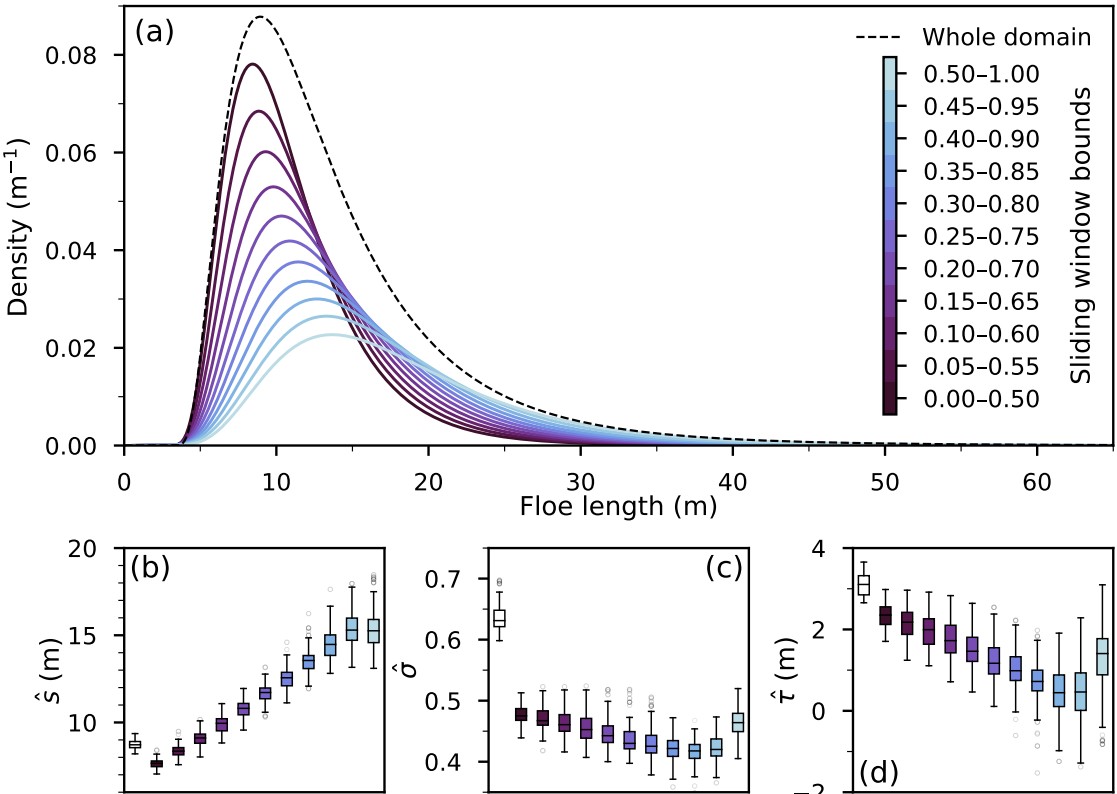

**Figure 6.** (a) FSDs, fitted to lognormal distributions, in the whole domain and various sub-domains, as described in the text. The area under the partial FSDs are scaled by the number of observations, so that summing the first and the last windows, covering non-overlapping halves of the domain, yields the whole domain FSD. The effective number of floes, relative to the total number of floes in the domain, decreases steadily from $0.64$ to $0.36$. Densities averaged over $500$ realisations. (b–d) Variation of the estimated parameters among the $500$ realisations. In each panel, the leftmost, white box corresponds to the whole domain; the coloured boxes follow the colour bar in panel (a).

$b_{\mathrm{sup}} \leq 1$, not on all floe lengths $\{L_j \mid j \in \{0, \ldots, N_f - 1\}\}$ but on the subset

$$\left\{ L_j \mid b_{\mathrm{inf}} < \frac{\Lambda_j}{\Lambda} \leq b_{\mathrm{sup}} \right\} \tag{28}$$

where $\Lambda_j = \sum_{m=0}^{j} L_m$ is the cumulated floe size up to floe $j$ and $\Lambda = \Lambda_{N_f - 1}$ is the total length of finite ice in the domain. We ignore the open water gaps in the definition of the breakup width. These monochromatic densities are then combined as detailed in Sect. 5.1. The difference $b_{\mathrm{sup}} - b_{\mathrm{inf}}$ gives the width of the sliding window, that we fix at $0.5$. The results of this

procedure, applied to the reference configuration introduced in Sec. 5.2, are presented in Fig. 6 (a).

The distributions remain remarkably lognormal, with the distribution parameters following regular trends for most positions of the sliding window (Fig. 6 (b–d)). Our breakup parametrisation generates floes that tend to get smaller as they get further

away from the semi-infinite floe marking the right boundary of the domain. This is true across all periods. As a consequence, the prevalence of larger floes grows for increasing $b_{\mathrm{inf}}$ values, shifting the distribution mode towards larger floes while thickening the distribution tail. This behaviour is similar to the effect of increasing the thickness, presented in Fig. 3 (b).

## 5.4 Forecast based on fitted parameters

We expand the analysis conducted in Sect. 5.2 to other combinations of $H_s$, $h$, $\varepsilon_c$. For simplicity, we focus here on investigating the effect of varying one variable at a time from their reference values. Some alternative behaviour may emerge from multivariate simulations, which are outside the scope of this study. Hence, when the value of one variable is specified, the other variables assume their reference values, stated in Sect. 5.2. We assess the suitability of fitting the lognormal model though exploratory analysis, as in Sect. 5.2. Histograms of these simulations are presented in Appendix C.

We observe a remarkably good fit over most of the parametric space explored, with a few notable limitations. The smallest waves (significant wave heights between $40\,\mathrm{cm}$ and $60\,\mathrm{cm}$) give rise to different patterns, which we do not analyse further. This is mostly due to them causing small amounts of breakup, leading to a limited number of floes. The empirical distributions obtained with thicker ice ($h > 1.4\,\mathrm{m}$) are less skewed, have a more pronounced peak and thinner tails than the fitted lognormals. Across most configurations, some limitations arise in the tails, with fits for stronger ice over-predicting larger floes, while fits for higher waves under-predict them. We report the estimated parameters, averaged over 500 realisations for each configuration, in Fig. 7 (a–i). We note in Fig.7 (g) that $\hat{\tau} < 0$ for large enough wave height, which would allow the model to generate negative floe sizes. This issue will be discussed in the next section. Additionally, we compute the Kolmogorov–Smirnov statistics to further qualify the goodness of fit. The evolution of this statistics is presented in Appendix D.

As can be seen in Fig. 7, the lognormal fit parameters have fairly simple dependences on the physical variables and can be interpolated between computed values, at the exception of the low-amplitude outliers (Fig. 7 (d)). We observe in Fig. 7 (k,l) that the mode of the FSD (see Eq. (27)), or modal floe size, grows with stronger ice, i.e. thicker floes or a larger strain threshold. This behaviour is analogous to the monochromatic case, as reported in Fig. 3. More surprisingly, the modal floe size first decreases with larger wave heights, reaching a local minimum for $H_s = 1.2\,\mathrm{m}$ before increasing with wave height. We believe this may emerge from the repartition of spectral energy between wave periods. Small significant wave heights do not induce much breakup, both because small-period waves, dominating the spectrum, are effectively reflected, and because the smaller amplitudes cause lower strains. On the contrary, higher significant wave heights have a larger component of high-period waves, leading to a smaller curvature of the ice floes, causing lower strains as well. This forecast mode has a strong dependence on the fitted shape parameter (Eq. (27)). The sample mode is not captured accurately by the fit for $H_s$ in the 1.4 to $2.2\,\mathrm{m}$ range, but the same behaviour is visible on the histograms (Fig. C1). As the peak propagating wavelength is proportional to the significant wave height, this non-monotonic evolution does not support the wavelength alone governs the dominant floe size, as has been conjectured and observed (Dumas-Lefebvre and Dumont, 2021; Herman et al., 2021).

The distribution parameters do not have a clear physical significance by themselves. Beyond the estimation of summary statistics, their main interest is the generation of floe sizes samples without the numerical cost of running the physical model. We illustrate such forecasts, and the associated errors, in Fig. 8. We use the mean distributions, parametrised by $\bar{\theta}$ for each

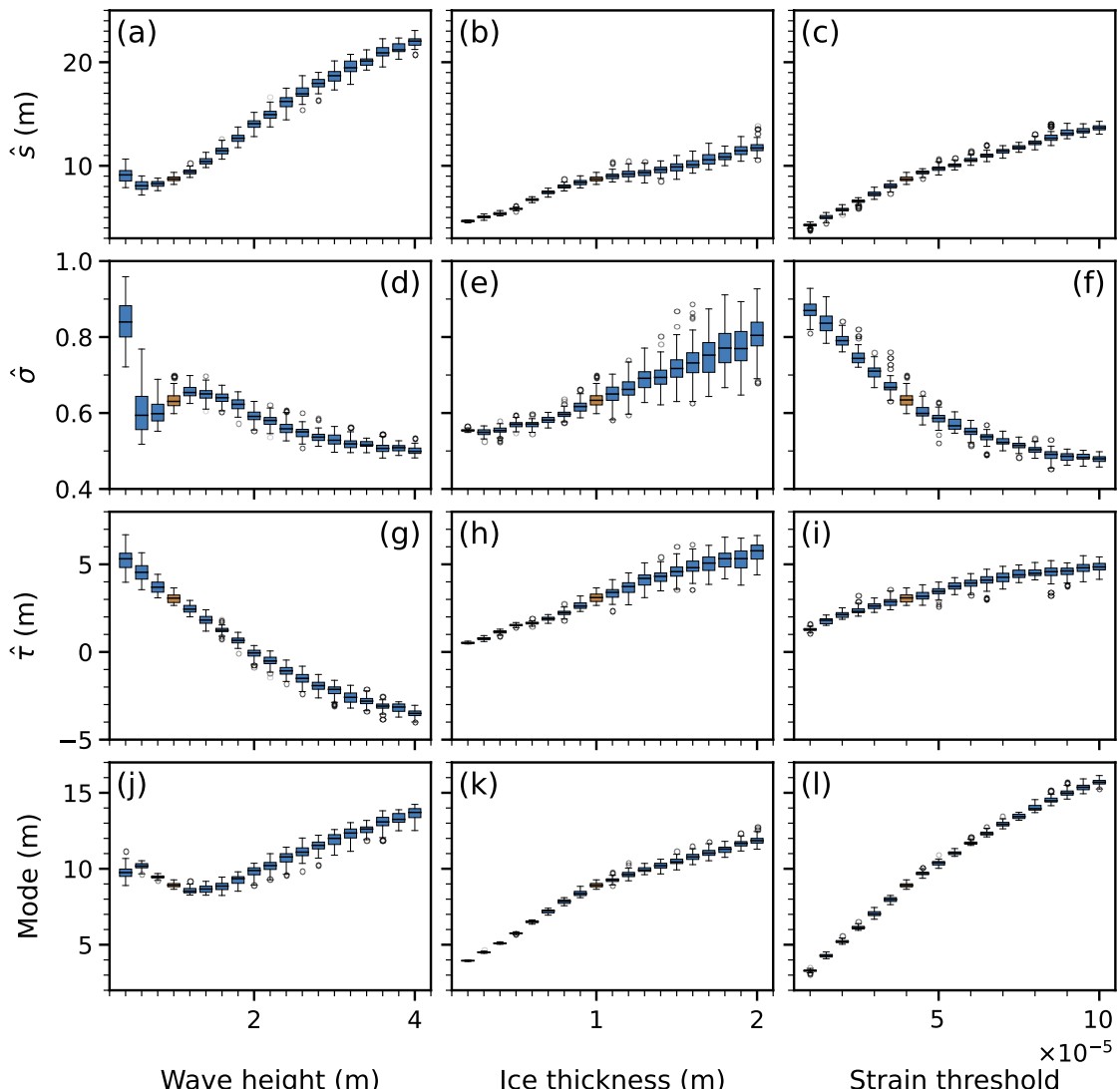

**Figure 7.** (a–i) Box-and-whisker plots of the estimated lognormal parameters for different model configurations. (j-l) Box-and-whisker plots of the modal floe size derived from the estimated parameters (Eq. (27)). The plots show the variation across 500 realisations. The reference configuration is highlighted with a contrasting colour, where the size of the boxes allow.

model realisation, to determine the ranges of floe sizes that would be predicted by the lognormal model. Every vertical slice in Fig. 8 (a–c) is a representation of the predicted FSD for the relevant physical variable on the horizontal axis.

Figure 8 (b,c) again show the dependence of the FSD on $h$ and $\varepsilon_c$ leads to a behaviour similar to the monochromatic case. 440    Increasing the ice thickness shifts the median floe size towards larger values and increases the spread, the distribution covering a larger span, especially for floes sizes beyond the 75th percentile. The smaller floe categories, below the 50th percentile, are

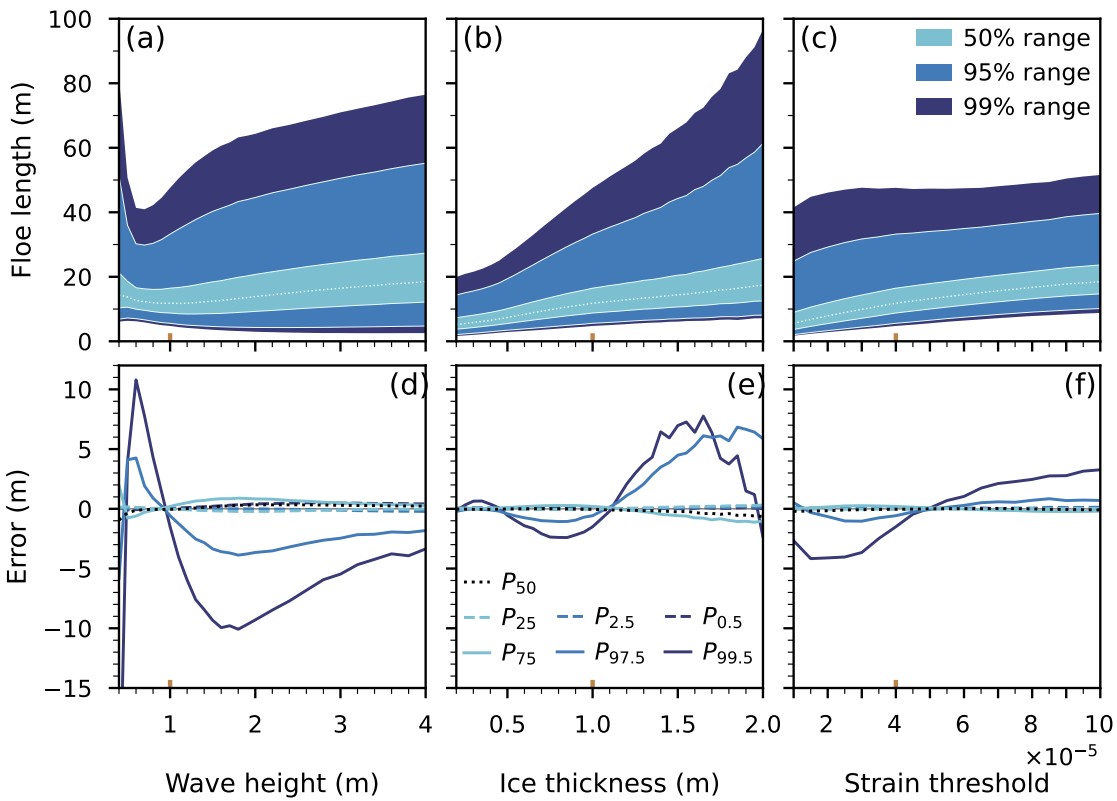

**Figure 8.** (a–c) Ranges of forecast distributions, generated with mean estimated parameters. The lightest colour area on each panel is the interquartile range; the dotted line denotes the median. The colour areas are symmetrical around the median. (d–f) Errors, with respect to the experimental quantiles, on the forecast quantiles. The represented quantiles bound the colour areas in panels (a–c). The 25th, 50th and 75th percentiles are respectively the first quartile, the median and the third quartile. Negative values suggest over-prediction, while positive values suggest under-prediction. The reference configuration is highlighted by coloured ticks on the horizontal axes.

shifted upwards as well and do not contribute to the increasing spread. Increasing the strain threshold sparks a steady increase in median floe size without much effect on the spread for $\varepsilon_c \geq 3 \times 10^{-5}$. The fraction of floes in the smallest size categories tends to increase. Increasing the significant wave height (hence, indirectly, the peak wavelength) leads to a more nuanced

behaviour. The general trend is an increase in spread for both small and large floe extremes. There seems to be an inflexion point around $H_s = 1.2\,\text{m}$ for the growth of the 99.5th percentile, this wave height corresponding to the mode local minimum (see Fig. 7 (j)).

Figure 8 (d–f) points out the limitation of the lognormal model by showing the differences between numerical results and statistical predictions for chosen percentiles. As expected, differences arise for more extreme quantiles, corresponding to the long right tail of the distribution. These differences are more pronounced for small wave height and large thickness

configurations. The prevalence of extreme floe sizes in the numerical results is, by definition, low. The errors on the three

quartiles, are smaller than $1\,\mathrm{m}$ on all the spans of the studied domains. The bulk of the FSD is hence well characterised by the lognormal model.

## 6 Discussion and conclusions

The emergence of a lognormal FSD from repeated wave-induced breakup is the key outcome of this paper. It contrasts with the power laws often assumed in modelling studies (Williams et al., 2013; Bennetts et al., 2017; Boutin et al., 2020), or with more narrow distributions from process-based sea ice breakup modelling (Herman, 2017; Montiel and Squire, 2017). Contrary to the assumption made by Williams et al. (2013), we systematically obtain floes larger than half the wavelength (monochromatic simulations) or half the peak wavelength (polychromatic simulations). Most floes (about $95\,\%$) are, however, smaller than that threshold. Anecdotally, the lognormal distribution has been reported for the size of brash ice pieces in navigation channels (Huang, 1988; Bonath et al., 2019). One of its earliest applications was the description of particles sizes from repeated fragmentation events (Kolmogoroff, 1941). In a companion paper, we attempt to make a connection between the repeated fragmentation theory of Kolmogorov and the breakup of a sea ice cover caused by a wave event (Montiel and Mokus, 2022).

The lognormal model does come with limitations. Field observations show the extensive spatial variability of the FSD. For instance, Paget et al. (2001) and Inoue (2004) both report an increase in the relative number of small floes when going towards the ice edge, respectively in the Antarctic and the Arctic. Our modelling results mirror this trend, highlighting the difficulty to settle on an all-around FSD parametrisation. We purposely ignore thermal and internal stress effects to focus on the effect of waves on the FSD, so validation with observational data would only be appropriate for a MIZ post wave-induced breakup. As detailed in Sect. 5.4 and illustrated in Fig. 8 (d–f), the distribution struggles to capture the behaviour of the most extreme floe sizes in our simulations. We note, however, that for such floe sizes wave-induced breakup is not likely to be the dominant mechanism governing the evolution of the FSD (Roach et al., 2018). We further found that the lognormal fit is not valid for small waves. The range of thickness we analysed is at the limit of validity, suggesting that even in this simplified setting the lognormal is not universal. Again, in such regimes, thermal and internal stress effects are likely to dominate over wave-induced breakup. Another point of concern is that we have $\hat{\tau} < 0$ for $H_s \geq 2\,\mathrm{m}$, meaning the probability of sampling negatively-sized floe is not 0. Constraining the MLE to yield positive estimates led to poor fit performance. This issue, which concerns a small fraction ($< 10^{-5}$) of the floes, can be easily circumvented by artificially bounding and rescaling the density function.

Our polychromatic forcing simulations (Sect. 5) lie on the underlying assumption that waves of different periods act independently on the ice. Although it is coherent with linear wave theory, which does not resolve non-linear interactions between wave components, it is unclear whether the resulting FSD describes reality appropriately. Other polychromatic parametrisations can be used to simulate the repeated breakup. For a given geometry, we could also compute the strain field obtained by linear superposition of strain fields at individual wave periods (with incident amplitude sampled from the prescribed spectrum and random phase). We compared the two approaches in the conference proceedings Mokus and Montiel (2022) and showed that they typically yield different results, with the strain superposition method generally leading to less breakup and larger floes. Qualitatively, however, the distributions follow similar shapes. It is not clear which one of these two approaches is phys-

ically more justifiable, as the strain-averaged approach assumes steady state to be reached by waves of all periods at the same time, even though dispersive effects tend to separate in time different spectral components of the sea state. Alternatively, the period-amplitude pairs can be sampled randomly between two model iterations (Montiel and Squire, 2017). Ultimately, the breakup of an ice cover in response to a wave event is a transient process. Consequently, validations of spectrum-generated FSDs will need to be sought against time-dependent simulations or experiments that control the wave forcing. Recent work by

Dumas-Lefebvre and Dumont (2021) and Passerotti et al. (2022) may provide the necessary datasets to conduct such validation studies.

The Pierson-Moskowitz spectrum chosen in our polychromatic simulations was selected for its simplicity. To make sure our results are not qualitatively sensitive to the choice of spectrum, we conducted additional simulations of FSD generation for a range of different spectra, including symmetric ones. The simulations are described and results are discussed in Appendix B.

In short, we find the emergence of right-skewed FSDs is consistent across the spectra considered, suggesting the lognormal FSD model is not an artefact of the choice of forcing spectrum.

Additionally to the wave height, the ice thickness and the strain threshold, we analysed the effect of varying the ice viscosity $\gamma$ (not shown here). We observed a significant contrast between $\gamma = 0$ (purely elastic ice) and $\gamma > 0$. The simulations we run with purely elastic floes are the only ones that reached the maximum number of iterations, set at 1000. It seems that

multiple scattering alone is not effective enough at attenuating wave energy, leading to a rapid and sustained growth of the number of floes. However, marginal differences exist between ensuing distributions as long as some viscosity is introduced ($\gamma$ in $1$–$100\,\mathrm{Pa\,s\,m^{-1}}$). Williams et al. (2013) used the same dissipation scheme with $\gamma = 13\,\mathrm{Pa\,s\,m^{-1}}$ derived from a 1979 campaign in the Bering Sea, while Mosig et al. (2015) fitted $\gamma = 6.9\,\mathrm{Pa\,s\,m^{-1}}$ to a 2012 Southern Ocean dataset (Kohout and Williams, 2015). Massom et al. (2018) derived $\gamma = 13.5\,\mathrm{Pa\,s\,m^{-1}}$ from the same dataset. We used $\gamma = 20\,\mathrm{Pa\,s\,m^{-1}}$ throughout,

which is a bit more conservative but, as stated, does not significantly impact the results. Although associated with the ice, this parameter can be thought of as a parametrisation of the collection of all wave dissipation effects (Squire and Montiel, 2016).

A framework to model the evolution of the ice thickness distribution (ITD) has been introduced by Thorndike et al. (1975). The ITD does not have a preferred functional form (Dupont et al., 2021) and is usually represented at the sub-grid level in sea ice model, such as CICE (Hunke et al., 2021), by various thickness categories. Horvat and Tziperman (2015) extended

this framework to include the floe size through a joint floe size and thickness distribution, which evolves under the action of separate physical processes such as thermodynamics, ridging and wave-induced breakup. Therefore, the authors did not have to make any assumption on the shape of the FSD. This method was adapted and ported to CICE (Roach et al., 2018) then coupled to a wave model (Roach et al., 2019). The model setup of Roach et al. (2019) was then used to train a neural network model in order to accelerate the source terms estimation in the FSD evolution model presented in Horvat and Roach (2022), which hence

aims to replicate it. Their formulation for wave-induced breakup relies on generating a discrete probability density of floe sizes, dependent on the wave field (controlling surface elevation) and floe thickness (controlling wave attenuation, when not handled by a wave model), by deriving strain from the attenuated wave field and using it to populate a histogram of floe sizes. However, any other way to generate that density may be substituted, including parametric forms. If more evidence were to point towards the lognormal being a compelling choice for FSDs observed in nature, with parameters that could be linked to combinations

of wave (significant wave height, peak period or wavelength) and ice (thickness) properties, as our study suggest they can, it would be a straightforward candidate. The (truncated) power law is another obvious candidate for such a substitution, and hindcasts comparing the two approaches could be a way to weigh in favour of one or the other.

Clauset et al. (2009) analysed 14 empirical datasets of different continuous variables, originally modelled with power laws, coming from a mixture of research areas. They observe that the power law is statistically appropriate for 8 of these datasets, 525 while the lognormal holds for 13 of them. They use a relative goodness of fit test to show that the lognormal is more suitable than the power law in 12 cases, and significantly so in 4 cases, concluding: "In general, we find that it is extremely difficult to tell the difference between log-normal and power-law behaviour" (Clauset et al., 2009). Stern et al. (2018) recommended a procedure for analysing floe size data in order to raise awareness of better fit methods, with considering an alternative distribution as an optional step. We believe this study heads in that direction and that the lognormal distribution should be 530 considered as a viable alternative. Revisiting some studies tabulated by Stern et al. (2018) with this hypothesis could shed some light on its validity. Such results are presented in Montiel and Mokus (2022).

Parametric distributions may never be flawless descriptions of the quantities they model. In this study, we show the relevance of the lognormal distribution when considering wave-induced breakup. We describe the evolution of the distribution shape for a range of ice properties, under various wave forcings. These results aim at being a step towards the parametrisation of wave 535 action in FSD-evolving models.

*Code and data availability.* The model code, software tools developed for analysis and the resulting preprocessed output presented in this paper are publicly available (Mokus and Montiel, 2021). The raw output is available upon request.

## Appendix A: Post-breakup floe positioning

This section details the process of redistributing the floes after a breakup event. The length of the gap between two floes does 540 not matter inasmuch as the fluid is inviscid and no wave energy is lost to it. The two main constraints are preserving the order of the floes, and ensuring they do not overlap.

Floes are positioned from left to right and localised by their left edge, whose location is drawn from a uniform distribution. The leftmost floe is placed in an interval of fixed width, symmetric around its location at the previous iteration. For subsequent floes, the left bound corresponds to the right edge of the last positioned floe (on their left). The right bound corresponds to the 545 previous right bound, augmented by the length of the last positioned floe. Location are drawn between these two bounds; an illustration is given in Fig. A1. As the width of that interval quickly tends to 0, we enforce a minimal length ($\delta_{min}$) for our random draw. It is set to $1\,\mathrm{cm}$ through the paper. It does not mean that floes cannot get arbitrarily close to one another, but that if so needed, the right bound is moved further right to enforce this minimum width. The room allocated to the first floe ($\delta_{init}$) is set to $100\,\mathrm{m}$ through the paper.

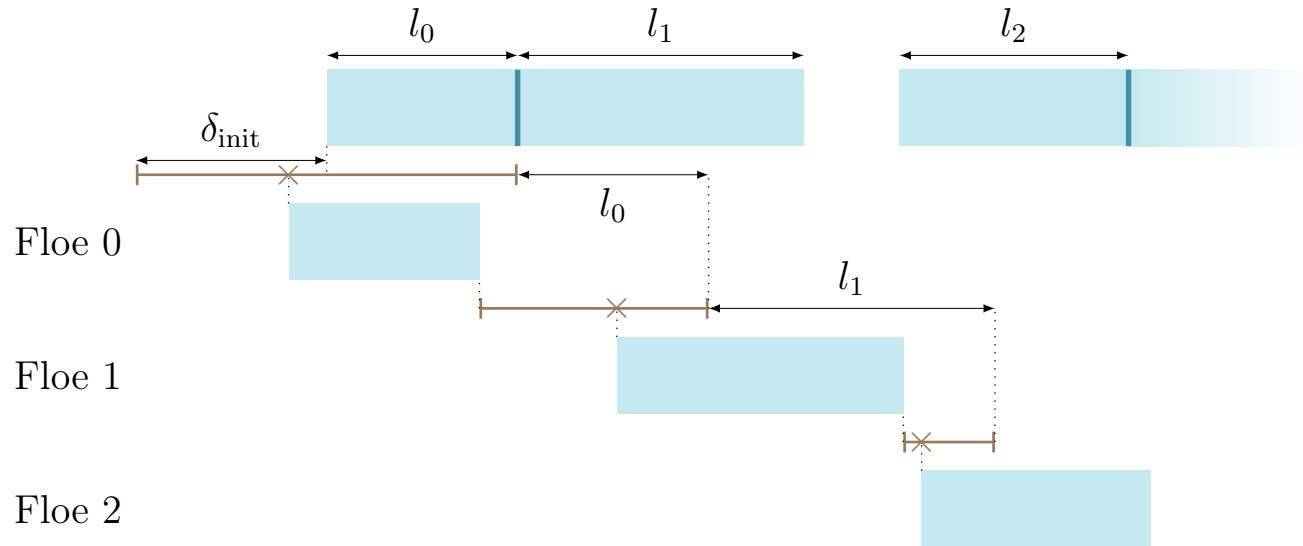

**Figure A1.** Illustration of the floe repositioning method. The top row shows current floes with identified breakup location marked by vertical bars and the resulting lengths. Successive rows show the iterative positioning as described in the text. Below each row, a segment shows the interval from which a location will be randomly drawn for the next floe; the cross marks that location.

**Table A1.** Distribution sensitivity to various parameters of random positioning. All quantity, except the sample size, are expressed in metres. The statistics are averages computed over 50 realisations. The support is truncated to the 0.5th and 99.5th percentiles.

| $\delta_{init}$ | $\delta_{min}$ | mean | std | sample size | support |
|---|---|---|---|---|---|
| 50 | 0.01 | 23.0 | 8.8 | 1053.6 | 10.9, 57.0 |
| 100 | 0.001 | 22.9 | 8.7 | 1039.6 | 10.7, 56.1 |
| 100 | 0.01 | 23.1 | 8.9 | 1007.9 | 10.8, 58.3 |
| 100 | 0.1 | 23.3 | 9.1 | 1057.1 | 10.8, 58.8 |
| 200 | 0.01 | 23.2 | 8.9 | 1046.9 | 10.7, 57.3 |

We ran simulation with alternative values to ensure these values do not have any impact on our results. We used the case presented in Sect. 4, with $T = 8\,\mathrm{s}$, $a = 50\,\mathrm{cm}$, $h = 1\,\mathrm{m}$, $\gamma = 20\,\mathrm{Pa\,s\,m^{-1}}$. Summary statistics and CDFs are presented in Table A1 and Fig. A2.

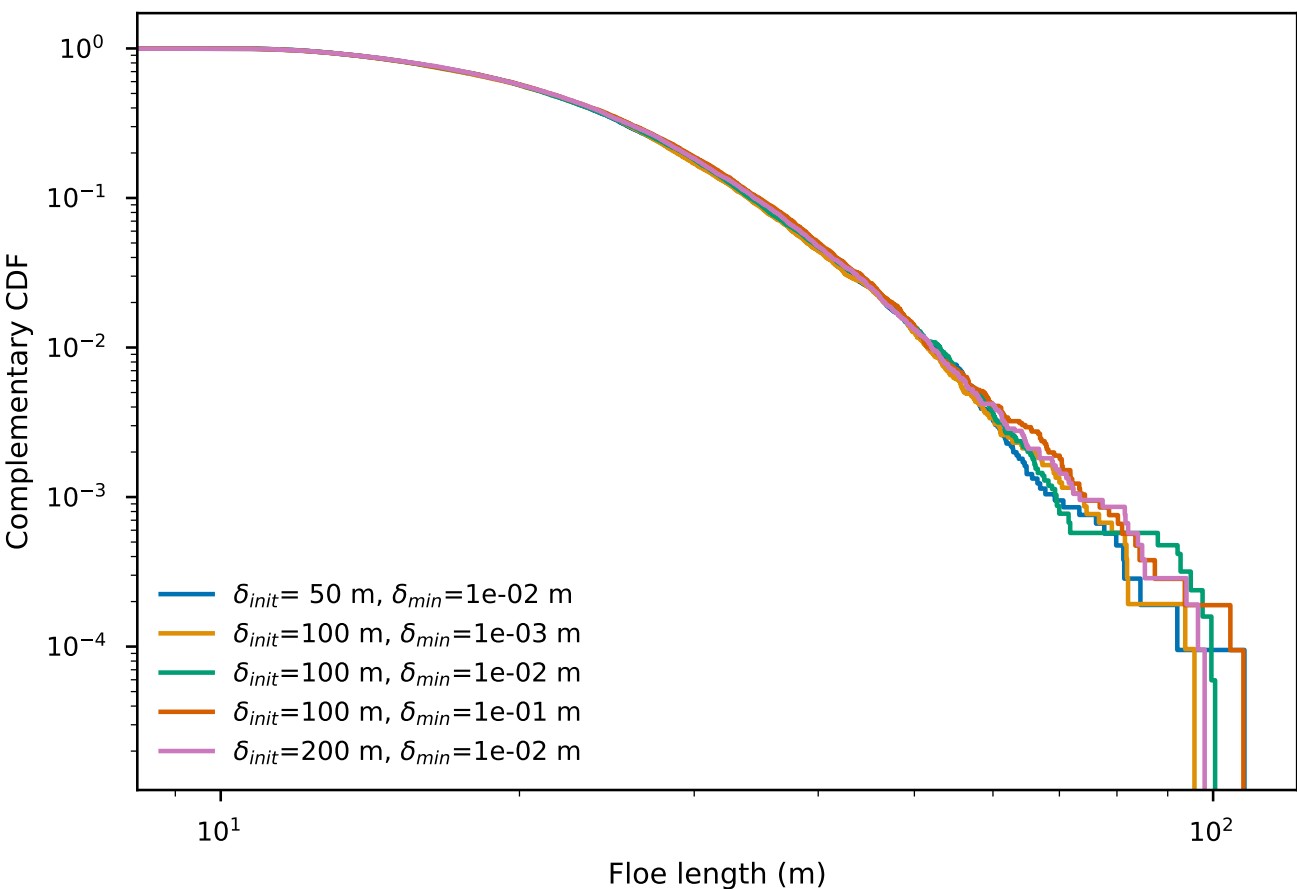

**Figure A2.** Complementary CDFs of floe lengths, averaged over 50 model realisations, with varied positioning parameters and model parameters mentioned in the text.

## Appendix B: Sensitivity to the choice of spectrum

We ran comparisons using alternative weighting functions, represented on Figure B1. Descriptions of these functions can be found in Table B1.

The lognormal density function has, qualitatively, a shape similar to the Pierson-Moskowitz spectrum expressed as a function of frequency. However, we obtain similar, skewed unimodal densities with a range of symmetrical weighting functions, as displayed on Figure B2.

The one case that stands out, with a lot of secondary peaks, is n_md. It corresponds to a function giving more weights to high frequency waves (5 to $15\,\mathrm{m}$), which is unrealistic in the context of our model.

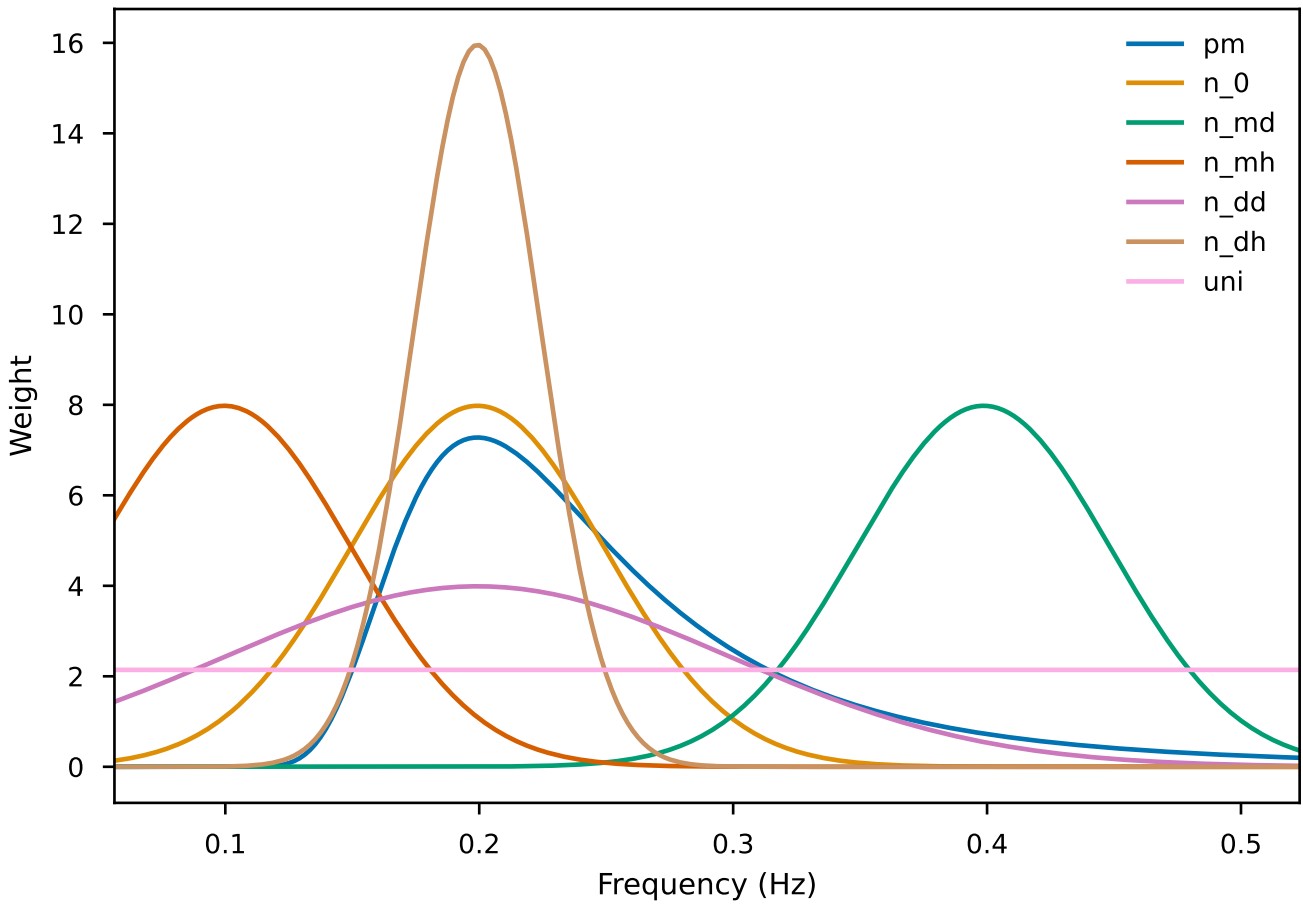

**Figure B1.** Representation of functions used as alternative weights. They are all normalised on the positive real half-line; because of the finite range of frequencies supported by our model, truncations imply some of them integrate to less than unity. Description of the legend entries can be found in Table B1.

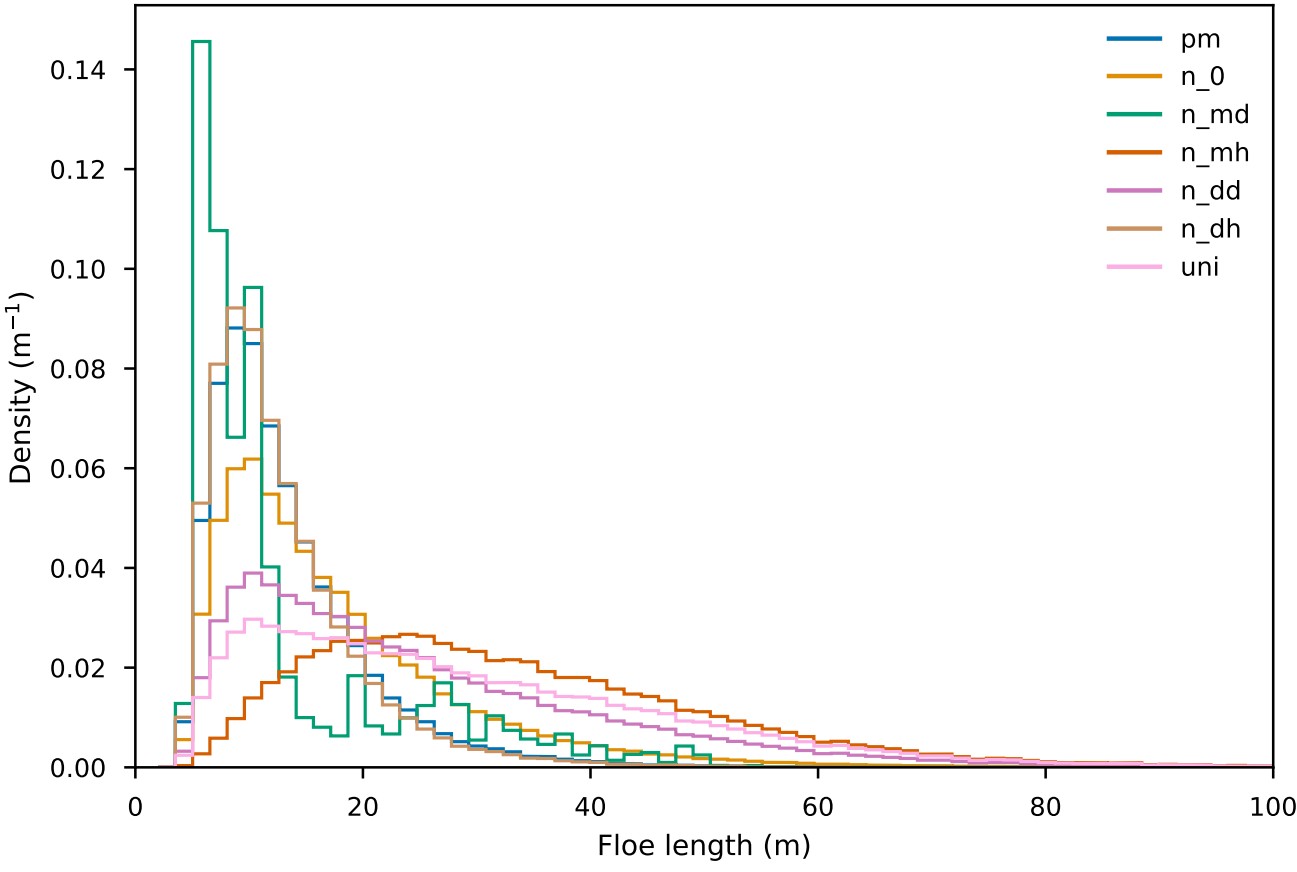

**Figure B2.** Densities obtained when combining monochromatic model runs with different weighting functions, as represented on Figure B1.

**Table B1.** Details of several weighting methods.

| Name | Type | Effective sample size |
|------|------|-----------------------|
| pm | Pierson-Moskowitz, $T_p = 5\,\text{s}$ | 23451 |
| n_0 | Gaussian, $\mu = 0.2\,\text{Hz}$, $\sigma = 0.05\,\text{Hz}$ | 42859 |
| n_md | Gaussian, $\mu = 0.4\,\text{Hz}$, $\sigma = 0.05\,\text{Hz}$ | 550 |
| n_mh | Gaussian, $\mu = 0.1\,\text{Hz}$, $\sigma = 0.05\,\text{Hz}$ | 65347 |
| n_dd | Gaussian, $\mu = 0.2\,\text{Hz}$, $\sigma = 0.1\,\text{Hz}$ | 71280 |
| n_dh | Gaussian, $\mu = 0.2\,\text{Hz}$, $\sigma = 0.025\,\text{Hz}$ | 19158 |
| uni | uniform | 77486 |

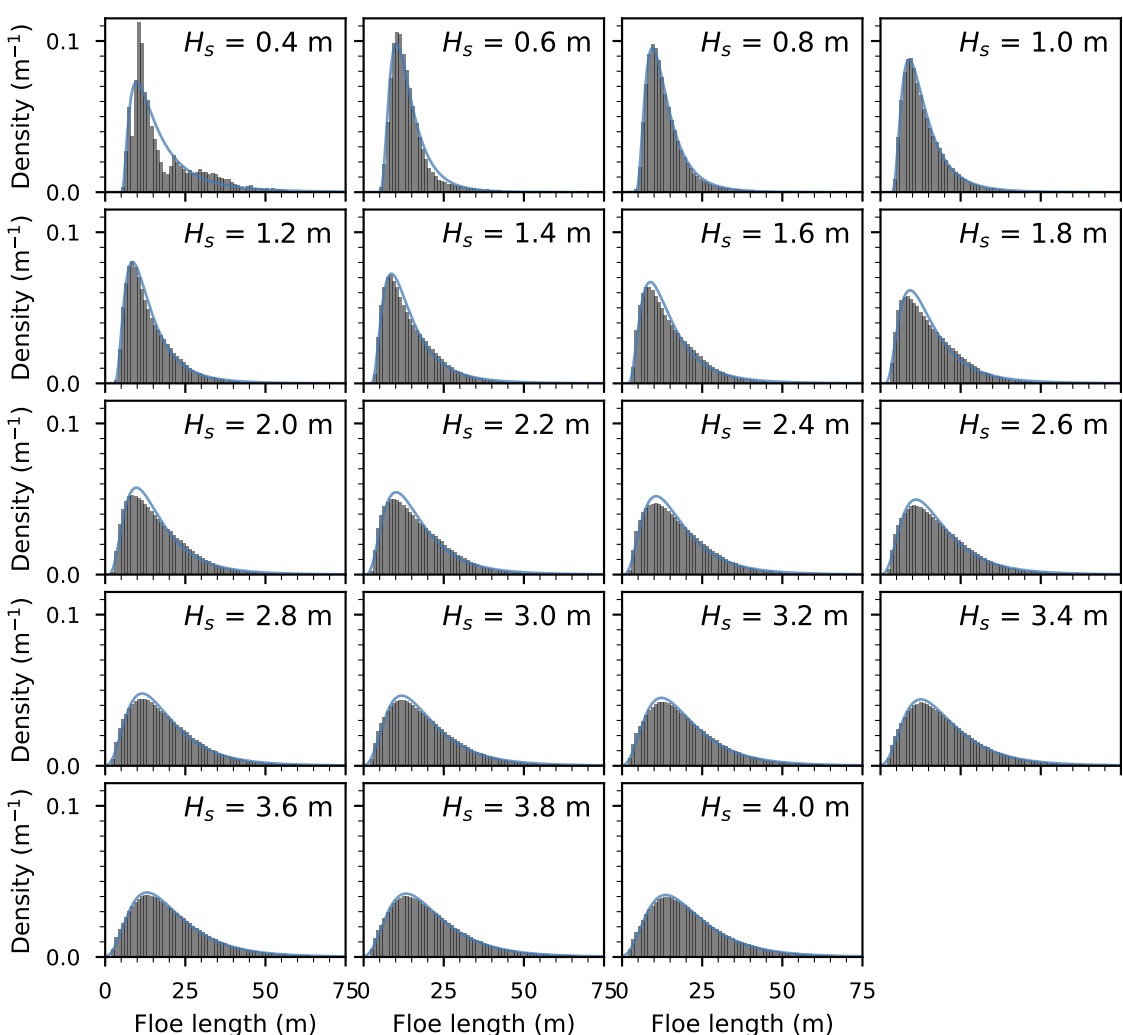

**Figure C1.** Histograms and average lognormal fits, as described in Sect. 5.2, for varying significant wave height.

## Appendix C: Histograms

## Appendix D: Kolmogorov–Smirnov statistics

The Kolmogorov–Smirnov statistics $D_{\mathrm{KS}}$ is the largest difference between an empirical cumulative distribution function (CDF) and a reference CDF (Massey, 1951). By definition of the CDF, $D_{\mathrm{KS}}$ is bounded by 0 and 1. When the distribution parameters have been estimated from the data, $D_{\mathrm{KS}}$ can be used to run a Lilliefors test (Lilliefors, 1967). By comparing $D_{\mathrm{KS}}$ to a critical value, depending on sample size and a chosen confidence level, one uses this test to reject a distribution hypothesis – not to confirm it. However, the power of this test, and others, notoriously increases with the sample size, making them able to

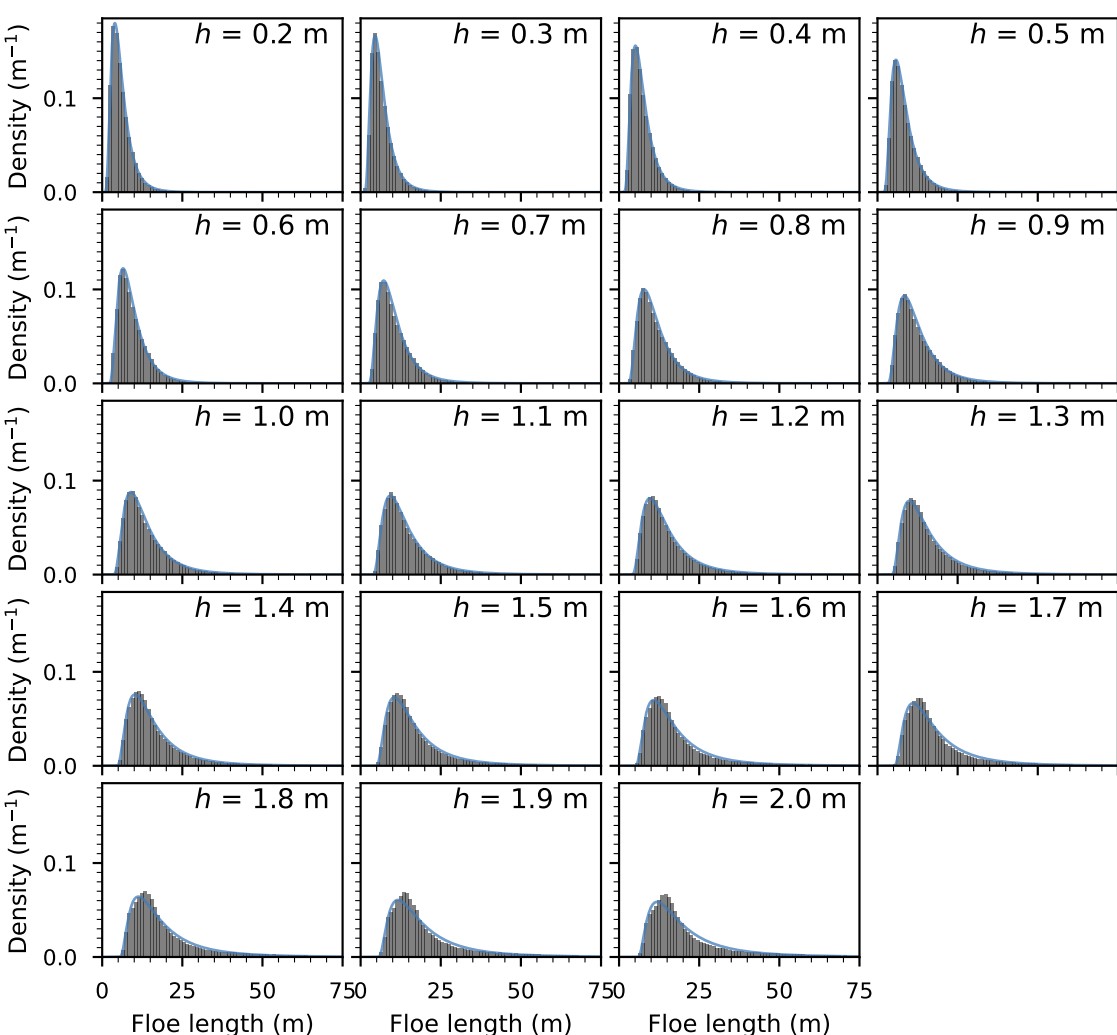

**Figure C2.** Histograms and average lognormal fits, as described in Sect. 5.2, for varying ice thickness.

detect trivial deviations from a reference distribution. This is a simple consequence of the fact that a model cannot perfectly fit the data. Hence, these tests only give a binary answer, not taking into account the usefulness of an imperfect model. A more

purposeful alternative consists in studying the relative goodness of fit between different models, which we do not explore in detail here.

Instead of rejecting the lognormal hypothesis at an arbitrarily-chosen confidence level, we report $D_{KS}$ as an indicative performance metric to compare our different configurations. More specifically, for each fitted lognormal with estimated parameters $\hat{\theta}$ (that is, 500 different realisations per model configuration), we generate a random sample of size $N_{eff}$ from the distribution.

We use Kish's effective sample size (Kish, 1965), the rounded up ratio of the squared sum of weights to the sum of the squared

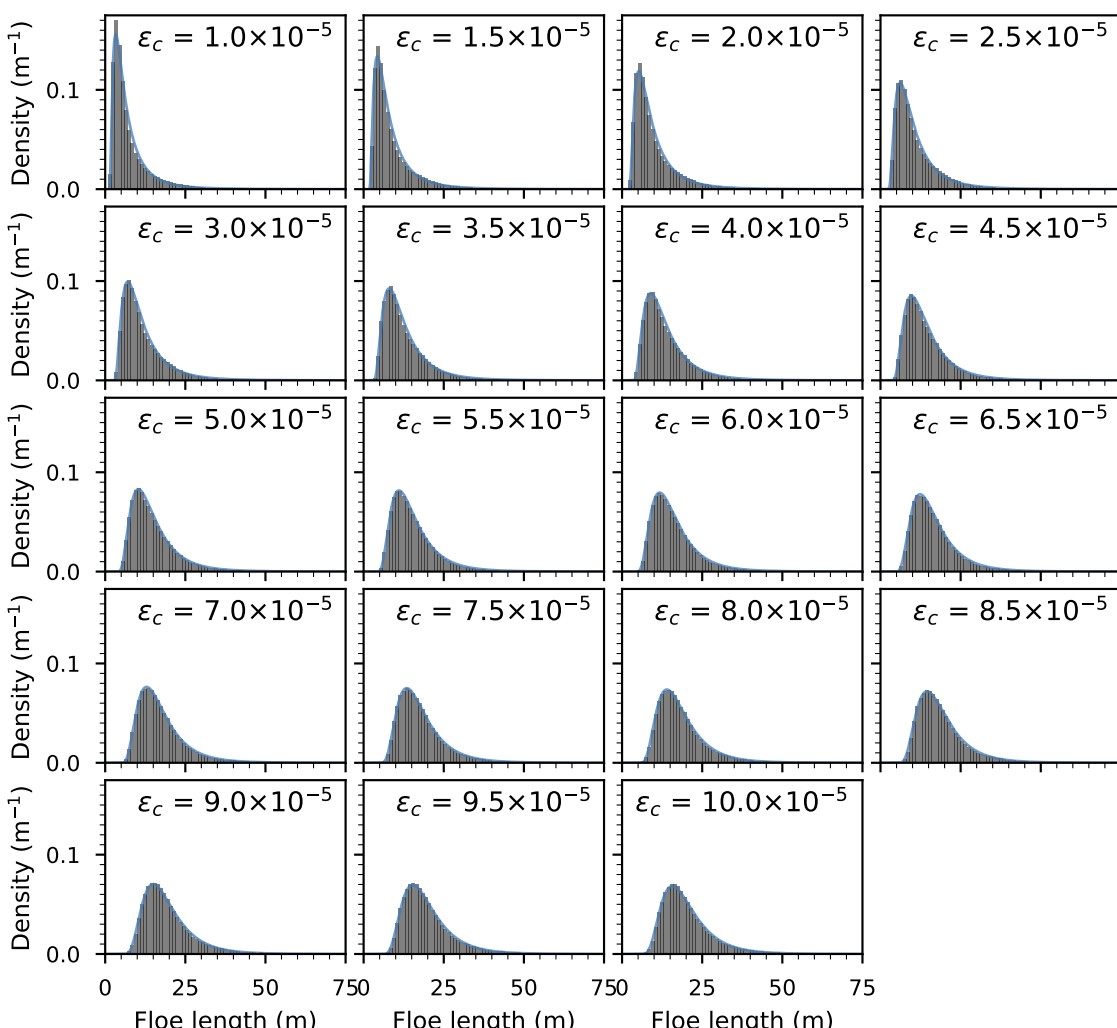

**Figure C3.** Histograms and average lognormal fits, as described in Sect. 5.2, for varying strain threshold.

weights, as $N_{\mathrm{eff}}$. We use maximum likelihood estimation (MLE) to estimate $\hat{\theta}_b$ from this sample, and we compute $D_{\mathrm{KS}}$ for this sample and the distribution parametrised by $\hat{\theta}_b$. We then define $\Delta D_{\mathrm{KS}}$ as the difference between $D_{\mathrm{KS}}$ from the random sample and $D_{\mathrm{KS}}$ from our data. We repeat these three steps 1000 times to derive the distribution of $\Delta_{\mathrm{KS}}$ for each model configuration. This is analogous to the bootstrapping method described by Clauset et al. (2009). It ensues that $\Delta D_{\mathrm{KS}}$ is bounded by $-1$ and $1$, with $\Delta D_{\mathrm{KS}} > 0$ indicating cases where our data is fitted by the lognormal model better than data actually lognormally sampled, in terms of distance between the CDFs.

We report the results on Figs. D1–D3. This procedure quantitatively illustrates the conclusions derived from analysing histograms and quantile–quantile plots.

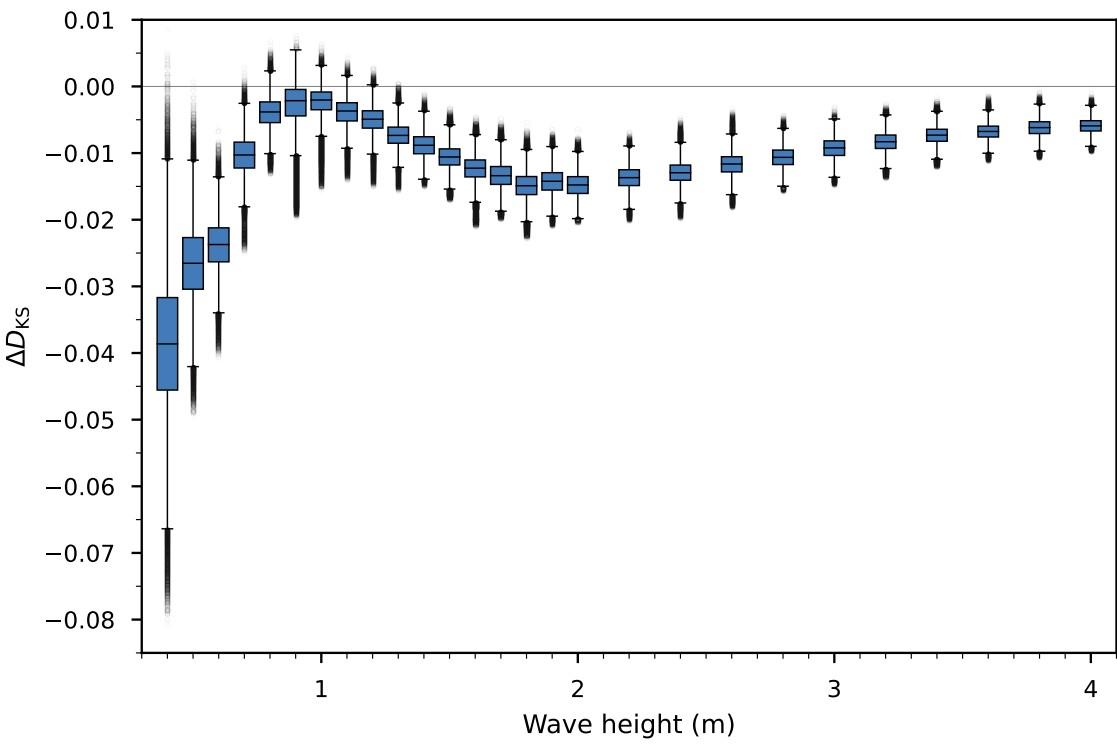

**Figure D1.** Distribution of $\Delta D_{\mathrm{KS}}$, as defined in the text, for varying significant wave height. The boxes are bounded by the first and third quartiles and the black lines are medians. The whiskers lengths is one and a half times the interquartile range. Black circles represent outliers.

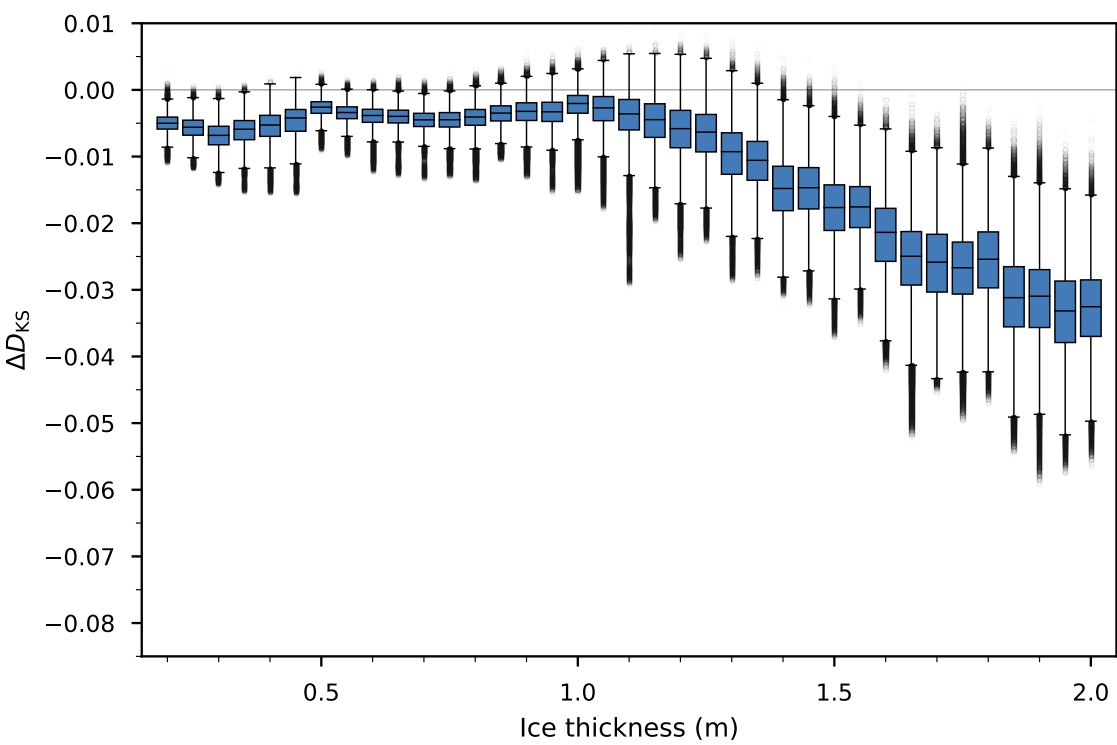

**Figure D2.** Same as Fig. D1 for varying ice thickness.

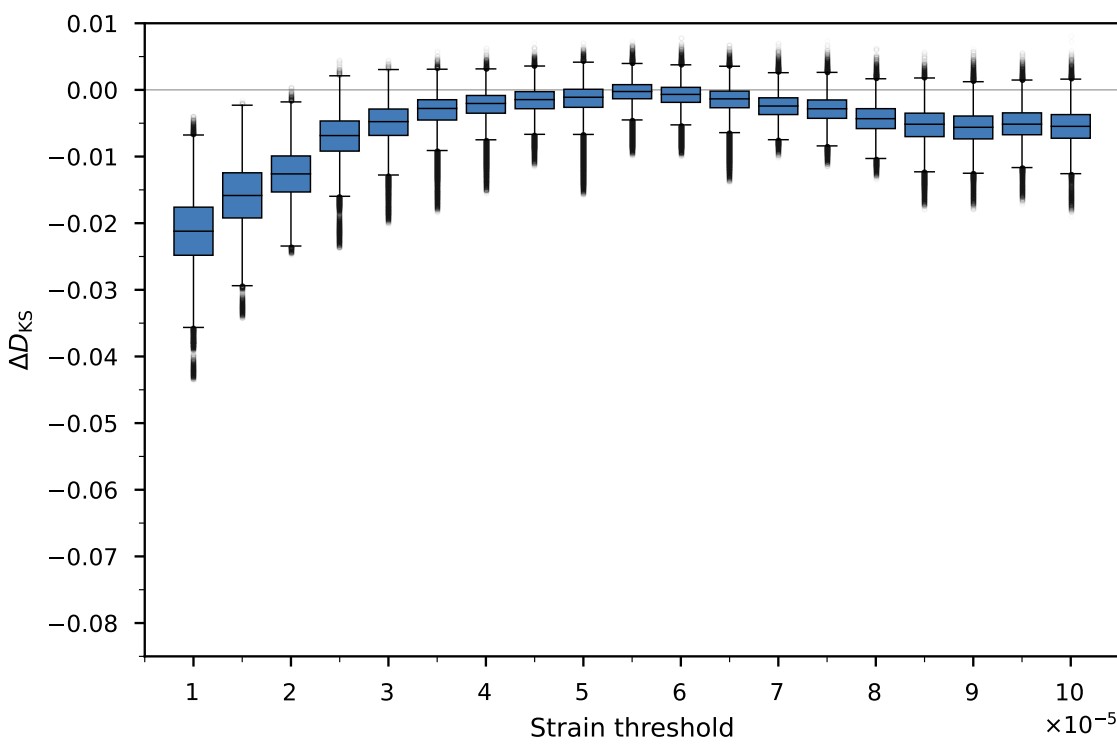

**Figure D3.** Same as Fig. D1 for varying strain threshold.

*Author contributions.* NM and FM designed the numerical experiments. NM developed the model code, ran the simulations and conducted the analysis. NM prepared the manuscript with significant inputs from and under the supervision of FM.

*Competing interests.* The authors declare that they have no conflict of interest.

*Acknowledgements.* The authors thank Christopher Horvat for proposing the method of combining FSDs obtained at individual frequencies in order to form the FSD resulting from the breakup by a wave spectrum (Eq. (23)). NM acknowledges the financial support from a University of Otago doctoral scholarship. This work was funded by the Marsden Fund, project 18-UOO-216, managed by Royal Society Te Apārangi. FM was also supported by the New Zealand's Antarctic Science Platform (Project 4) and Royal Society Te Apārangi (Marsden project 20-UOO-173).

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
