# Peer review of "Wave-triggered breakup in the marginal ice zone generates lognormal floe size distributions: a simulation study"

_The Cryosphere, 2021_

## Community Comment (CC1)

**Review of "Wave-Triggered breakup in the marginal ice zone generates lognormal floe size distribution" by Mokus & Montiel (2021)**

This very well written paper presents a 2-D hydrodynamic model for wave-induced sea ice breakup which combines linear wave theory and viscoelastic sea ice rheology in order to compute the scattering of wave by sea ice floes. By using an empirical strain threshold to define the floe size resulting from breakup, the FSD resulting from breakup follows a lognormal distribution under realistic wave forcings thus demonstrating that a preferential size is indeed generated by the process. They also show that the median floe size evolves with both wave period and ice thickness, a result that partly contrasts with the findings of Fox and Squire (1991) and Herman (2017) in which the FSD is independent of the sea state. I'd recommend a consideration of the comments below before the article is published.

**Introduction :**

- This section is a very good review of the FSD topic that gives the reader a broad view of the subject and helps understand what the challenges related to its observation and its inclusion in models are. Lines 35 to 40 sum up the problem with the power-law very efficiently.

**Preliminaries :**

- This section poses the framework very clearly and efficiently. Figure 1 is great and really contributes to the understanding of the model throughout the reading of the paper.

**Methods :**

- The description of the scattering model is clear and exceptionally well made.

- Regarding the breakup parametrization, using argmax $\varepsilon_j$ as the position of fracture is in a sense arbitrary since the floe could break anywhere between the positions where $\varepsilon_c$ and $\varepsilon_{\max}$ are reached. How sensitive is the resulting FSD and its statistical moments to the position of the breakup ?

- What is the relationship between the fracture position and sea ice properties $(h, Y, \gamma)$ ? And what about wave properties $(a, T, \lambda)$ ? Say $x^*$ is the position where breakup happens, is it possible to obtain functional relationships $x^* = f(h, Y, \gamma, a, T, \lambda)$ with your data? Such information would be helpful for the translation of your results into larger scale models. Knowing the shape of the FSD is a great step but being able to circumscribe it with physical properties would bring an even more complete physically-based parametrization.

- Figure 2 illustrates well and concisely the algorithm.

- In table 1, a value of 6 GPa for sea ice Young's modulus is displayed. Where does this value come from and how is the FSD affected by it ?

**Monochromatic forcing :**

- Figure 3 is great to get grasp the physics of the problem as it is formulated in the paper but one key aspect regarding strain is missing in my opinion. To be more specific, how is the strain distributed spatially in floes and how does that evolve in time ?

**Main comment :** The principal concern I have with this paper is on how the FSD is built since this could have heavy repercussions on its shape and thus on the title of the paper. There are indeed many ways to compute the FSD, which are toroughly analyzed by Stern et al. (2018), and none is really better than another. But, depending on what it the goal of the paper is, which seems here to be "[aimed] at being a step towards parametrization of wave-action in FSD-evolving models", a particular way of computing the FSD can be advantageous. The mention of "histogram of the length" in the caption of figure 3 makes me believe that the probabilities of each floes are computed using the frequency of observation in the model. Dumas-Lefebvre and Dumont (2021) have shown that using the frequency of observation of the floes to build the FSD may lead to a bias on the estimation of the modal size and incidentally on the shape of the FSD. We have proposed using the partial concentration, which is the definition of both the ITD (Hunke and Lipscomb, 2010) and more recently the FSD (Bateson et al., 2020) used in global sea ice models, rather than the frequency of observation to compute the probability densities of each floe size category. With this framework, we have obtained a FSD that i) has a significantly different shape than with the frequency of observation approach, ii) a distribution mode that better corresponds with what can be seen visually in observations and iii) a FSD that is directly translatable to larger scale models.

With that in mind, could you describe how the FSD is obtained ? Secondly, I strongly suggest to re-compute your FSDs with the partial concentration approach since it could have an impact on the shape of the distribution and would then alter the title of the paper. For the mathematical details of the computation, I refer you to Dumas-Lefebvre and Dumont (2021) and if you have any questions, do not hesitate to reach out to me.

**Forecast based on fitted parameters**

- Figure 7 does a great job at showing how the parameter vector and modal floe size evolve relative to wave height, ice thickness and strain threshold. Can your model give insight on how are the waves and sea ice properties respectively responsible of the modal floe size, minimal and maximal sizes as well as the spread of the distribution ?

**Discussion and conclusions :**

- The discussion at lines 405 to 413 is good but to be more complete, line 414 should include Dumas-Lefebvre and Dumont (2021) since we provide data on a "post wave-induced breakup" FSD.

**References**

Bateson, A. W., Feltham, D. L., Schröder, D., Hosekova, L., Ridley, J. K., and Aksenov, Y. (2020). Impact of floe size distribution on seasonal fragmentation and melt of Arctic sea ice. *The Cryosphere Discussions*.

Dumas-Lefebvre, E. and Dumont, D. (2021). Aerial observations of sea ice break-up by ship waves. *The Cryosphere Discussions*, pages 1–26.

Fox, C. and Squire, V. A. (1991). Strain in shore fast ice due to incoming ocean waves and swell. *Journal of Geophysical Research*.

Herman, A. (2017). Wave-induced stress and breaking of sea ice in a coupled hydrodynamic discrete-element wave-ice model. *Cryosphere*.

Hunke, E. C. and Lipscomb, W. H. (2010). CICE : the Los Alamos Sea Ice Model Documentation and Software User ' s Manual LA-CC-06-012. *Research Report*, pages 1–76.

Stern, H. L., Schweiger, A. J., Zhang, J., and Steele, M. (2018). On reconciling disparate studies of the sea-ice floe size distribution. *Elementa*.

---

## Author Comment (AC1)

**1 General comments**

The authors use a 2D wave–ice model involving wave scattering, viscoelastic dissipation and a strain breaking threshold to conduct a detailed statistical analysis of the steady-state FSD produced by wave forcing. Strong evidence is given that the model predicts lognormal FSDs. The study is communicated clearly and the key outcome is potentially a valuable contribution towards modelling the marginal ice zone.

I recommend revisions before publication.

We thank the reviewer for their positive comments and suggestions, which are addressed below.

**2 Specific comments**

**2.1 Introduction**

The Introduction is missing an overview of the considerable literature on modelling wave propagation in the MIZ. At present, readers could be led into thinking that the model used is accepted by the community, when, as the authors surely know, debate and open questions remain. There are, for example, different methods for modelling wave scattering and many different models of viscous damping. Certain models have been validated using experimental data. A similar comment applies to models of ice breakup caused by waves. It should be clear at the end of the overview why the particular wave propagation and ice breakup models have been chosen for the present investigation.

We acknowledge that our literature review was biased towards models similar to that used in our study. We will add references and a discussion regarding other methods, such as continuous viscous layer and 3D wave scattering models.

**2.2 Page 2**

The two paragraphs starting from the bottom of page 2 are not particularly relevant for the study presented (e.g. the ideas are not picked up again later) and would be better in Sect 6, leading into a discussion on how the proposed model and findings could be implemented in CICE, etc. Sect 6 would also be strengthened by comments on possible implications of the reduced dimension of the model (e.g., in comparison to the 3D model

of Montiel and Squire 2017) and whether the predicted FSD properties are consistent with the ideas used by Dumont, Williams and co to parameterize power-law FSDs (such as the maximum floe size being half a wavelength).

We thank the reviewer for this suggestion which would strengthen the discussion section. We will incorporate it in the revised manuscript.

**2.3 Sect 5.1**

At the beginning of Sect 5.1, the move from monochromatic to polychromatic forcing requires more explanation and justification. Presumably the definition of the FSD for polychromatic forcing in equation (23) is computationally efficient, but is it representative of the ensemble average FSD created by (random) irregular wave forcing that obeys the prescribed spectrum? Can examples be given to demonstrate this? Better understanding of this aspect of the model will improve interpretation of the results. Incidentally, I was unable to find $f_L$ and $\tilde{f}_L$ when scanning back through the paper at this point. Perhaps the latter could be introduced in Sect 2.

We considered an alternative way to introduce the polychromatic forcing, with results shown in the conference proceeding paper Mokus and Montiel (2022). Instead of considering the weighted average of FSDs from monochromatic forcings, we considered the FSD resulting from the repeated breakup by an irregular-wave-induced strain field simulated from a discretised spectrum with random phases. The results are quantitatively different, but the distribution shape remains similar (unimodal, mode clearly distinct from the smallest observation), with the lognormal model significantly stronger than the power law model. An illustration of the difference can be seen on Figure 1. However, we do not think one of these two parametrisations of a polychromatic forcing is obviously better. Both rely on different sets of assumptions. In particular, the irregular wave-forcing simulation assumes steady state to be reached by waves of all periods at the same time, even though longer waves propagate faster. Our initial approach assumes different periods act independently to break the ice, and that their effects can be averaged over. Which one is physically the most sensible is unclear and will need experimental confrontation.

We will insist on the assumptions and underlying limitations in the revised manuscript, and mention the alternative parametrisation (strain superposition) in the discussion.

[Figure]

Figure 1: Comparison of results from two ways of considering the polychromatic forcing. Lognormal fits overlayed over histograms. The leftmost histogram (orange hue) corresponds to the method presented in Section 5 of the present paper. The rightmost histogram (blue hue) corresponds to the alternative method presented by Mokus and Montiel (2022), where we use strain superposition to determine the fracture points. Both histogram areas are normalised, the log x-axis skewing this perception.

**2.4 Title**

A title that indicates the scope of the study would be better, e.g. Model predictions of lognormal floe size distributions in the marginal ice zone caused by wave forcing

We will consider alternative titles.

**3 Minor comments**

**3.1 25**

With thinner and weaker first-year ice becoming dominant in the Arctic

'in the Arctic' added

**3.2   28**

Elaborate on the sentence starting The individual description.

We mean that the dynamic of every floe, at the basin scale, cannot be reasonably determined and kept track of. We will clarify this in the revised manuscript.

**3.3   55**

The sentence on short time scales for breakup appears to contradict the steady state model assumption.

Breakup happens on time scales shorter than thermodynamics processes, that our model does not resolve. Recent observations (Dumas-Lefebvre and Dumont 2021) showed the breakup front moves slower than the wave front within the ice cover, so we believe these assumptions hold. We will clarify the sentence in the revised manuscript.

**3.4   Sec 3.1**

Similar wave scattering models should be referenced at the beginning of the section Kohout and Meylan 2008; Montiel et al. 2012, and any notable differences identified.

Our model is indeed directly inspired by these. We will make the connection to other similar models more obvious in our description.

**3.5   149**

travelling and evanescent …

We will add a reference to evanescent modes.

**3.6   170**

> For completeness, say that the complex roots can become purely imaginary for high frequencies and/or thick ice.

> We believe this point is slightly out of scope, as this is very unlikely to happen in any geophysically realistic setting (Bennetts 2007).

**3.7 178**

> I think the phases are used to normalize rather than cancel out the exponential terms.

> We meant cancel in the sense of making them neutral with respect to multiplication. We will make the phrasing clearer.

**3.8 Eqn (13)**

> Replace the full stop with a comma.

> Corrected.

**3.9 248+250**

> for every floe and none of the floes break

> Corrected.

**3.10 253**

> Give the distribution used to randomly redistribute the floes after breakup.

> Floes are positioned from left to right and localised by their left edge. The leftmost floe is placed at a random location, drawn from a uniform distribution, around its location at the previous iteration. For subsequent floes, the left bound corresponds to the right edge of the last positioned floe (on their left). The right bound corresponds to the previous right bound, augmented by the length of the last positioned floe. A location is drawn from a uniform distribution between these two bounds; an illustration is given in Figure 2. As the width of that interval quickly tends to 0, we enforce a minimal length for our random draw. Floes can still get arbitrarily close to one another, as long as they do not overlap. The room allocated

[Figure]

Figure 2: Illustration of the floe repositioning method. The top row shows current floes with identified breakup location marked by vertical bars and the resulting lengths. Successive rows show the iterative positioning as described in the text. Below each row, a segment shows the interval from which a location will be randomly drawn; the cross marks that location.

to the first floe (labelled $\delta_{init}$ on the schematic in Figure 2) as well as that minimum width are set to $100\,\mathrm{m}$ and $1\,\mathrm{cm}$, respectively. We ran simulation with alternative values to ensure these values do not have any impact on our results.

We will include these details as an appendix to the revised manuscript.

**3.11   258**

Give details on the local resonances plus references.

For any single realisation of the array, local resonances can take place due to additive interference between scattered waves. These can be filtered out through ensemble averaging. This behaviour, and the solution, are described by Kohout and Meylan (2008). The reference will be added in the revised manuscript.

**3.12   Figure 3d**

The levelling off/decrease of the median floe size with increasing ice thickness for T=8s is interesting and worth discussing in the text.

We will discuss this feature in the revised manuscript.

**3.13   Figure 4 caption**

Figure 4 caption: State the amplitude(s) used.

It is the same as in the previous figure, $50\,\mathrm{cm}$. We will correct the omission.

**3.14   348**

348: Space needed after the full stop.

Corrected.

**3.15   428**

Note that the value $\gamma = 13.5\,\mathrm{Pa\,s\,m^{-1}}$ was derived from measurements in the Antarctic MIZ (Massom et al. 2018).

We thank the reviewer for this reference, that we did not know about. A smaller value ($6.9\,\mathrm{Pa\,s\,m^{-1}}$) is derived in Mosig et al. (2015). The value $13\,\mathrm{Pa\,s\,m^{-1}}$ is used in Williams et al. (2013) and subsequent studies; however, it is unclear how this parameter may depend on, e.g., the ice thickness or rigidity, so we settled on a slightly more conservative estimate. Even though not presented here, we conducted experiments with a range of viscosities. We will develop this point in the discussion, and add the suggested reference to our review.

**3.16**

Mathematics needs a capital M in the institution name.

Corrected.

**References**

Bennetts, L. G. (2007). "Wave scattering by ice sheets of varying thickness". PhD thesis. University of Reading.

Dumas-Lefebvre, E. and D. Dumont (2021). "Aerial observations of sea ice break-up by ship waves". In: *The Cryosphere Discussions* 2021, pp. 1–26. DOI: 10.5194/tc-2021-328.

Kohout, A. L. and M. H. Meylan (Sept. 2008). "An elastic plate model for wave attenuation and ice floe breaking in the marginal ice zone". In: *Journal of Geophysical Research* 113.C9. DOI: 10.1029/2007jc004434.

Massom, R. A., T. A. Scambos, L. G. Bennetts, P. Reid, V. A. Squire, and S. E. Stammerjohn (June 2018). "Antarctic ice shelf disintegration triggered by sea ice loss and ocean swell". In: *Nature* 558.7710, pp. 383–389. DOI: 10.1038/s41586-018-0212-1.

Mokus, N. and F. Montiel (Apr. 2022). "Floe size distributions in irregular sea states". In: *Proceedings of the 37th International Workshop on Water Waves and Floating Bodies*.

Montiel, F., L. Bennetts, and V. Squire (Jan. 2012). "The transient response of floating elastic plates to wavemaker forcing in two dimensions". In: *Journal of Fluids and Structures* 28, pp. 416–433. DOI: 10.1016/j.jfluidstructs.2011.10.007.

Montiel, F. and V. A. Squire (Oct. 2017). "Modelling wave-induced sea ice break-up in the marginal ice zone". In: *Proceedings of the Royal Society A: Mathematical, Physical and Engineering Science* 473.2206, p. 20170258. DOI: 10.1098/rspa.2017.0258.

Mosig, J. E. M., F. Montiel, and V. A. Squire (Sept. 2015). "Comparison of viscoelastic-type models for ocean wave attenuation in ice-covered seas". In: *Journal of Geophysical Research: Oceans* 120.9, pp. 6072–6090. DOI: 10.1002/2015jc010881.

Williams, T. D., L. G. Bennetts, V. A. Squire, D. Dumont, and L. Bertino (Nov. 2013). "Wave–ice interactions in the marginal ice zone. Part 2: Numerical implementation and sensitivity studies along 1D transects of the ocean surface". In: *Ocean Modelling* 71, pp. 92–101. DOI: 10.1016/j.ocemod.2013.05.011.

---

## Author Comment (AC2)

**1 General comments**

The manuscript „Wave-triggered breakup in the marginal ice zone…” by Nicolas Mokus and Fabien Montiel describes a numerical study of wave propagation in sea ice and wave-induced sea ice breaking. The main focus of the paper are the properties of floe size distributions (FSDs) resulting from breaking of ice with different properties (strength, thickness) by waves of different periods and amplitudes.

Undoubtedly, the problems discussed in the study are important for the current research on sea ice–wave interactions. Our better understanding of the physical mechanisms underlying wave-induced sea ice breaking is crucial for developing better parameterizations of those processes for large-scale sea ice and climate models. Although I find the manuscript and the results interesting and valuable, and the model developed by the Authors well presented, I have some doubts, described below, regarding some parts of the analysis. I recommend the manuscript for publication in The Cryosphere after a major revision.

We thank the reviewer for their positive comments and suggestions, which are addressed below.

**2 Major comments**

**2.1**

The main point in my critics is related to the procedure described in Section 5.1: the whole algorithm is based on an assumption that the FSD resulting from sea ice breaking on irregular waves is the 'weighted average of distributions resulting from monochromatic model runs'. Why?

I really can't see the reason why it should be so simple.

Let's consider a very simple example of a wave field composed of two monochromatic waves with very different wavelengths, and let's assume that wave #1 does break the ice and produces very small floes, and wave #2 is very long and doesn't break the ice at all (or produces very large floes). The ice sheet in that case would break into small floes, corresponding to that resulting from wave #1 anyway, so computing FSD from a weighted average would produce truly weird results!

It's the part of the spectrum that leads to breakup that's important, not the whole spectrum!

As the Authors rightfully demonstrate in their manuscript for monochromatic waves, the relationships between floe size, ice properties,

and wave length are quite complex and nonlinear, so there is no reason why the FSD resulting from a wave energy spectrum should behave as the Authors assume.

This is a valid concern. Obviously, this is a model, hence a simplification of reality. We will make clearer in our manuscript that our approach in Sect. 5 is indeed heavily assumption-dependent. The underlying assumption is that waves, carrying the whole energy of the spectrum at discrete time periods, would act independently to break the ice. The resulting distribution would be the average of the distributions generated by these independent periods, with weights related to the spectrum, in coherence with linear theory. In reality, waves of different periods would interact with one another, but our model cannot represent these interactions.

I have the impression that the shapes of FSDs in Fig. 6 to a large degree simply reflect the shape of the wave frequency spectrum, and that this is an artefact of the algorithm (or, more precisely, its part related to the computation of weighted averages).

In my opinion, it is a very weak part of the analysis, but the Authors don't even discuss those weaknesses.

Of course, as I have serious objections regarding the above-mentioned assumption, I have also doubts regarding the results presented in sections 5.2–5.4 of the manuscript.

We ran comparisons using alternative weighting functions, represented on Figure 1. Descriptions of these functions can be found in Table 1.

The lognormal density function has, qualitatively, a shape similar to the Pierson-Moskowitz spectrum when displayed as as a function of frequency. However, we obtain similar, skewed unimodal densities with a range of symmetrical weighting functions, as displayed on Figure 2.

The one case that stands out, with several secondary peaks, is n_md. It corresponds to a function giving more weights to high frequency waves (5 to 15 m), which is unrealistic in the context of our model.

Why can't the model be forced by a superposition of monochromatic waves? The scattering model is linear, isn't it, so it shouldn't be difficult. All one needs to do is to add up the wave solutions for individual spectral components (assuming random phases) and use those to compute strain (as, e.g., in section 6 of Kohout and Meylan (2008)).

[Figure]

Figure 1: Representation of functions used as alternative weights. They are all normalised on the positive real half-line; because of the finite range of frequencies supported by our model, truncations imply some of them integrate to less than unity. Description of the legend entries can be found in Table 1.

We implemented the approach suggested by the referee (Mokus and Montiel 2022, conference proceedings). The results are different (Figure 3), which is not an argument in favour of either of the methods. The later approach has limitations has well, as it assumes steady state to be reached by waves of all periods at the same time, even though longer waves propagate faster. These models would need experimental validation for additional support.

We will insist on the assumptions and underlying limitations in the revised manuscript, and mention the alternative parametrisation (strain superposition) in the discussion.

**2.2**

Are the FSDs obtained for monochromatic waves lognormal as well?

Why is that pdf introduced first in Section 5.2 and not earlier? That would allow comparisons between FSDs obtained for regular and irregular

[Figure]

Figure 2: Densities obtained when combining monochromatic model runs with different weighting functions, as represented on Figure 1.

wave forcing.

The PDFs obtained for monochromatic simulations seem to be well fitted by lognormal distribution as well (see Figure 4). This will be discussed in the revised manuscript. We delayed the introduction of the lognormal PDF to Sect. 5 of the manuscript for two reasons. First, Sect. 4 of the manuscript was meant to be an introduction to its Sect. 5. Therefore, we did not want to deep dive into a parametric representation of these FSDs. Second, the linear combination of lognormal PDFs (as presented in Sect. 5) is not, in general, a lognormal PDF. The opposite can be true, as a lognormal PDF can arise from the sum of other functions.

**2.3**

The algorithm, as described in Section 3.3, does not take into account the time evolution of breakup – in the sense that the breaking events during one "sweep" are all taking place at the same time instance, and a breaking event at one location does not influence what is going on in an immediate vicinity of that location (sudden stress release etc.).

Table 1: Details of several weighting methods.

| Name | Type | Effective sample size |
|------|------|----------------------|
| pm | Pierson-Moskowitz, $T_p = 5\,\text{s}$ | 23451 |
| n_0 | Gaussian, $\mu = 0.2\,\text{Hz}$, $\sigma = 0.05\,\text{Hz}$ | 42859 |
| n_md | Gaussian, $\mu = 0.4\,\text{Hz}$, $\sigma = 0.05\,\text{Hz}$ | 550 |
| n_mh | Gaussian, $\mu = 0.1\,\text{Hz}$, $\sigma = 0.05\,\text{Hz}$ | 65347 |
| n_dd | Gaussian, $\mu = 0.2\,\text{Hz}$, $\sigma = 0.1\,\text{Hz}$ | 71280 |
| n_dh | Gaussian, $\mu = 0.2\,\text{Hz}$, $\sigma = 0.025\,\text{Hz}$ | 19158 |
| uni | uniform | 77486 |

[Figure]

Figure 3: Comparison of results from two ways of considering the poly-chromatic forcing. Lognormal fits overlayed over histograms. The leftmost histogram (orange hue) corresponds to the method presented in Section 5 of the present paper. The rightmost histogram (blue hue) corresponds to the alternative method presented by Mokus and Montiel (2022), where we use strain superposition to determine the fracture points. Both histogram areas are normalised, the log x-axis skewing this perception.

[Figure]

Figure 4: Histograms from Fig. 3 of the manuscript, and associated lognormal fits.

> I'm not criticizing it, I just wonder whether/how this limitation can influence the resulting FSDs. What is the Authors' opinion about that?

> This is hard to answer. Wave scattering being solved in the frequency domain is an obvious limitation of our model, on which we will insist more in the revised manuscript. A truly transient model would be necessary to evaluate the influence of this simplification, and that is out of the scope of the present manuscript. In a sense, our iterative approach, allowing for each floe to break several time, is a (restricted) substitute for true time evolution.

**2.4**

> Figure 4a,b shows the total number of floes for various combinations of the model forcing. How does the width of the MIZ (i.e., the total length of the broken ice) change? It is an important parameter for several reasons, so it would be interesting to see plots analogous to those in Fig.4ab, but showing the MIZ width. Or at least some comments on that in the text.

> We thank the reviewer for this suggestion. We will modify Fig. 4 in the manuscript to show the MIZ width, as shown in Figure 5. The peak of MIZ width as a function of wavelength is reached at a wavelength slightly larger than that causing the largest number of broken floes. If weakening the ice increases the MIZ width, making it thicker, counterintuitively, does so as well.

[Figure]

Figure 5: Figure 4 from the manuscript with additional panels showing the width of the MIZ.

**2.5**

[Figure]

Figure 6: Attenuation of the extremum strain along a model transect, and associated exponential fit. The xaxis is the ordinal floe number in the propagation direction.

I know it's beyond the scope of this paper, but I'm just curious: Have the Authors analyzed the shape of the attenuation curves produced by their model? Are they approximately exponential, or are there deviations from the exponential curve (as in eq. 2.1 of Squire, Phil Trans A, 2018), especially close to the ice edge?

We have not done so consistently for all our tests. We did, at a preliminary stage, study the shape of the strain attenuation. These were indeed exponential (Figure 6). We may add a short discussion of wave attenuation in Sect 4.

**3 Minor, technical and other comments**

**3.1 Line 38**

'Hence…' suggests this sentence follows from the previous one, but I don't really see the connection. I think I know what is meant here, but I'd suggest formulating it more clearly.

We mean that the role waves played in the emergence of an observed FSD is not well established for many of these studies. We will alter the phrasing to make it clearer.

**3.2 Lines 41–43**

I'd suggest to add here that this technique not only leads to erroneous values of the power law exponents, but, in the first place, suggests the existence of power law tails even when there aren't any and when the pdfs aren't heavy-tailed at all.

We will make that addition.

**3.3 Line 93**

The recent paper by Dumas-Lefebvre and Dumont (currently under discussion in TCD: https://tc.copernicus.org/preprints/tc-2021-328/) is worth citing here, as it describes a wonderful observational dataset of sea ice breaking by waves. (It's not self-advertisement, I'm not an author of that paper.)

We agree. The paper is a more detailed version of Dumas-Lefebvre and Dumont (2020), that we cite. We will make the substitution.

**3.4 Lines 256–258**

I understand that those tests suggest that the details of how the floes are placed after breaking are not important.
Maybe it's a naïve question, but are those empty spaces between floes necessary? Does the algorithm work for densely packed ice field, with zero spaces between floes?

We consider the fluid to be inviscid, so the distance travelled by the waves between floes does cause attenuation. The numerical model would need a bit of tweaking for zero spacing to work, but currently spacing could be as small as machine precision allows (double precision floating numbers). We could have settled on assigning a random phase to the forcing wave, at each algorithm iteration, instead of randomly positioning the floes, to achieve the same result.

**3.5   Lines 265–266**

'FSD dispersion'. Dispersion? As the term 'dispersion' has a clearly defined meaning in the context of waves, I'd suggest replacing it here with 'median floe size'.

'Dispersion' also has a clearly defined meaning in the context of statistics. Variance and interquartile range are typical measures of dispersion. It is indeed confusing and so we will replace the word by e.g. 'spread'.

**3.6   Lines 268–269**

'a positive relationship between the ice mechanical resistance […] and the presence of larger floes'. But the skewness is larger for smaller strength and thinner ice, isn't it? The presence of larger floes itself can result from a simple shift of the distribution to the right and is not directly related to the skewness, so this sentence is a bit misleading.

The distributions being skewed is a general, qualitative comment and we do not quantify skewness in the paper. The presence of larger floes can indeed result from a shift of the distribution to the right; we do not mean to link it to the skewness in any way. We will alter the phrasing to make it clearer.

We did compute skewness here for the sake of the discussion (Figure 7). We do not observe any trends for stronger ice at higher time periods (from $T = 8\,\text{s}$); however, skewness does increase for weaker ice at shorter time periods ($T = 4\,\text{s}$). Interestingly, at a given $\varepsilon_c$, it does not evolve monotonically with $T$. The evolution of skewness as a function of ice thickness is not monotonic at any period, and we do not attempt to analyse it further.

**3.7   Lines 268–269**

[Figure]

Figure 7: (a) Sample skewness for different forcing periods and evolving strain threshold. The dots correspond to mean values, and the shaded area to one standard deviation around it, over 100 realisations. (b) Same as (a) for evolving ice thickness. In (a), $h = 1$; in (b), $\varepsilon_c = 4 \times 10^{-5}$. In both, $a = 50 \, \text{cm}$.

> And further: "Qualitatively, increasing $\varepsilon_c$ has only a moderate effect on the FSD and seems to be only affecting its mode, shifting it towards larger floes, while its shape remains the same." Is it really so? Are the shape parameters of the pdfs in Fig.3a really so similar? My impression from the figure is quite different. It might be wrong, of course, but please back up this statement by some numbers, e.g., skewness values (maybe you could add them to the panels in Fig.3a,b for those three cases presented?).
>
> As far as the mode is concerned, in Fig. 3a it changes by 100% between case 1 and 3, so I'd say it is a quite substantial change.

> The skewness can be seen as a measure of shape. As shown in Figure 7 (a), the skewness is indeed quite stable, for $T = 8 \, \text{s}$ and increasing $\varepsilon_c$. For reference, the skewness values corresponding to the histogram presented in Fig. 3 (a, b) of the manuscript are given in Table 2.
>
> The modes in Fig. 3 (a) of the manuscript do increase with $\varepsilon_c$, and we point it out. It can result from a horizontal translation of the distribution. It is a noticeable quantitative change, but not in contradiction with the shape evolving little. We will clarify that statement in the revised manuscript.

**3.8 Line 274**

Table 2: Skewness values for the different model configurations presented in Fig. 3 (a, b) of the manuscript. The given values are the means over 100 realisations and the associated standard deviations.

| $\varepsilon_c$ | $h$ | skewness |
|---|---|---|
| $2 \times 10^{-5}$ | $1\,\mathrm{m}$ | $1.48 \pm 0.15$ |
| $4 \times 10^{-5}$ | $0.5\,\mathrm{m}$ | $2.92 \pm 0.62$ |
| $4 \times 10^{-5}$ | $1\,\mathrm{m}$ | $1.46 \pm 0.23$ |
| $4 \times 10^{-5}$ | $2\,\mathrm{m}$ | $2.63 \pm 0.69$ |
| $8 \times 10^{-5}$ | $1\,\mathrm{m}$ | $1.63 \pm 0.37$ |

Line 274: 'the dispersion in floe sizes': again, it's not clear what exactly is meant here. The range of floe sizes? (i.e. pdf width?)

We mean the spread of the distribution, as represented in Figure 3 (c,d) by the interquartile ranges. We will change it in relation with comment 3.5.

**3.9  Line 277**

Line 277: crisp -> sharp? rapid?

Changed to rapid.

**3.10  Line 349**

Line 349: 'the definition of the ice edge is not clear, as it is period-dependent'. I don't understand this statement, please clarify. And further: 'the total length of ice in each period category'. Period category? Overall, I'd recommend rephrasing this whole paragraph, as it contains a lot of statements that are hard to follow (although the overall meaning is clear, of course).

By period category, we meant the period bins used to discretise the spectrum. The ice edge location, or the MIZ width, will depend on the wave period used to force the system in a particular monochromatic run. We agree on this paragraph needing some rephrasing.

**3.11  Lines 426–427**

Lines 426-427: 'scattering alone is not effective enough at dissipating wave energy'!!! Scattering does not dissipate energy at all! Moreover, in a 1D setting, scattering alone does not lead to wave energy attenuation within sea ice: even for an extremely long ice cover, the wave energy at its downwave end must be equal to the energy of the incomming wave minus the energy reflected from the from the upwave edge. In other words, if any attenuation is observed in the scattering-only model runs, it only results from numerical inaccuracies.

'Attenuating' is indeed better suited here than 'dissipating'. We believe it is clear from the context this comment is about multiple scattering. Successive reflections do lower the downwave amplitude, as expected by comment 2.5.

**References**

Dumas-Lefebvre, E. and D. Dumont (Feb. 2020). "Aerial observations of sea ice breakup by ship-induced waves". en. In: *ArcticNet Annual Scientific Meeting.* DOI: 10.13140/RG.2.2.23493.40164.

Kohout, A. L. and M. H. Meylan (Sept. 2008). "An elastic plate model for wave attenuation and ice floe breaking in the marginal ice zone". In: *Journal of Geophysical Research* 113.C9. DOI: 10.1029/2007jc004434.

Mokus, N. and F. Montiel (Apr. 2022). "Floe size distributions in irregular sea states". In: *Proceedings of the 37th International Workshop on Water Waves and Floating Bodies.*

---

## Author Comment (AC3)

**1 General comments**

This paper aims to develop an efficient model of wave breakup of sea ice floes including a random component of floe positioning that can be used to generate statistical descriptions of floe size (probability) distributions (FSD) that might emerge from wave breakup from sea ice and rapidly explore relevant parameter spaces within this setup (e.g. wave period, sea ice thickness). The study finds that the emergent FSD can be best characterised using a lognormal distribution and discusses implications of these results for finding the best fit to observations of floe size and for future parametrisations of floe breakup by waves in sea ice models. This work intersects two areas of research that have had significant focus in recent years: modelling the role of individual processes in determining the emergent FSD in sea ice models and modelling interactions between waves and sea ice and how sea ice can impact wave propagation. This study builds on earlier efforts to develop simple but accurate models of wave breakup of floes. The value of the model presented here is that it is efficient and can be used to rapidly explore relevant parameter spaces and include stochastic elements within the model to represent uncertainty / variability (in this case to capture variability in floe positioning without a full treatment of sea ice dynamics). I therefore believe this paper makes a useful contribution to both the sea ice and wave modelling communities, and also has potential value in understanding and characterising observations of floe size.

The scientific quality of the work presented is generally strong, with good associated analysis and discussion. The figures are of a very good quality and appropriate to the discussion. The structure of the paper seems fine and is easy to follow, though it would be good to see a more thorough overview of the paper structure at the end of the introduction. The paper reads well, is clear in its conclusions, and also has a representative abstract and title. I do have a couple of major concerns that would need to be addressed before I can recommend publishing. Firstly, I am not sure the methodology used has been sufficiently justified. Specifically, the choice to use monochromatic model runs and then taking the weighted average to determine the emergent FSD from a full wave spectrum is not properly justified / supported as a reasonable approximation. In addition, the study repeatedly refers to whether observations of the FSD should be fitted to a power law. Whilst this is an important discussion, I find the paper focuses too much on this point and insufficiently on other impacts / conclusions of the findings presented. Full details of these concerns are provided in the specific comments.

Overall, I believe that this paper is within the scope of The Cryosphere and, provided the above concerns can be adequately addressed, merits

publishing.

We thank the reviewer for their positive comments and suggestions, which are addressed below.

**2 Specific comments**

**2.1 General point**

The study uses this result to backup conclusions from other studies such as Stern et al. 2018 that other possible fits should be tested against observations of floe size, not just a power law. These conclusions are justified on the basis of the evidence presented. However, throughout the manuscript the authors question the validity of power law fits to FSD data. Whilst this is a reasonable and justified question to ask and one several previous papers have discussed as noted in the manuscript, I find this point is too frequently made within the manuscript, at the expense of other important results that emerge from this study, given this study does not appear to present any new evidence to suggest that a power law does not produce a valid fit to observed FSDs (as opposed to new evidence to support the testing of alternative fits to observations, which the study does present, as noted above). Even in regions of high wave activity, observed FSDs are not necessarily solely a result of wave breakup. Even if they are, there are physical features that may determine the FSD not considered within the model used here (e.g. variable ice thickness, existing weaknesses in the sea ice, fractures that are not perpendicular to the direction of wave propagation). The emergence of a lognormal distribution from this model does not necessarily tell us anything about the validity of a power law fit to observations of floe size unless this model can be validated using observations of an FSD under wave control, which has not been presented in this study.

We agree that the FSD is impacted by more than just wave activity. We have shown in a separate paper (Montiel and Mokus 2022, manuscript under revision) that the lognormal model can be applied to observations previously analysed under the power law hypothesis. However, wave conditions at the time of these measurements were not available. Confronting our results to more controlled experiments would be the only way to validate them. We will make more obvious that our findings do not negate the collection of studies based on the FSD following a power law distribution.

We acknowledge that any parametric model would be a simplification, and that many non-controlled factors, such as pre-existing cracks, can locally impact the FSD. Additionally, limits on the floe sizes that can be remotely discriminated tend to truncate observations. We note that the

tail of the lognormal distribution, excluding the region of values smaller than the mode, asymptotically identifies to a power law distribution.

**2.2 P2 L28–29**

'The individual description of these, floating pieces of sea ice is not possible.' What do you mean by this comment? Individual pieces of ice cannot presently be simulated in continuum models, but they can in discrete element models of sea ice.

We mean they cannot be represented at the scale of an ocean-wide numerical simulation. We will clarify this point in the revised manuscript.

**2.3 P3 L65–68**

You should also describe/discuss the most recent study from Horvat and Roach 2022 that introduced a machine-learning-derived parameterization of wave breakup of floes that can be used within the prognostic model.

We were not aware of this study when we submitted ours. We will include it in our revised manuscript.

**2.4 P2 L57–P3 L81**

In this section you have described existing treatments of wave breakup of floes within sea ice models but there are other approaches that you have not described e.g. both Bateson et al. 2020; Boutin et al. 2021 include treatments of wave breakup of sea ice within FSD models. It would be helpful to either briefly discuss these treatments or at least highlight that your discussion is not exhaustive.

We will add these references.

**2.5 P3 L80–81**

'Nevertheless, the model sensitivity analysis conducted by (Zhang et al. 2016) revealed compelling improvement on ice extent simulation when considering their FSD formulation.' What were the improvements? This statement is vague and should be clarified.

Their implementation yielded improvements in terms of ice extent and location of the ice edge. Their simulations including the parametrisation were better able to replicate observations than those which did not. We will make this point clearer in the revised manuscript.

**2.6 P4 L97–105**

It would be helpful to describe the overall structure of the paper at the end of the introduction i.e. describe how the paper proceeds, section by section.

We will add this description to the introduction.

**2.7 P8 L202–203**

Why did you decide to use a fixed sea ice thickness in your simulations? Do you anticipate that a lognormal distribution would still emerge if the sea ice thickness was variable in a single evaluation of the model?

This stems from the experiment design. Our scattering formulation assumes individual floes are of constant thickness. As we populate the MIZ by breaking off floes off a single initial floe, it comes out that all resulting floes share this initial thickness. This thickness could be altered after breakup has happened, but we choose to keep it constant as breakup happens on time scales shorter than those of the processes that would alter the thickness. As stated in the text, the model is capable of handling floes with varying properties, including the thickness. However, such simulations are outside the scope of this paper, that focuses on FSDs resulting from repeated breakup events. Therefore, we cannot comment on the FSDs we would obtain in such a simulation.

**2.8 P8 L205–206**

'A sensitivity analysis (not shown here) proved $N_v = 2$ to be adequate in terms of convergence.' Please provide more details on this. How are you assessing adequate convergence here?

We studied the convergence in terms of energy conservation. We placed five floes of finite length and zero viscosity in the domain. Scattering does not dissipate energy, so energy carried by travelling modes is expected to be conserved for reflection/transmission by individual floes and overall, when considering the array of floes as a single scatterer. Note that under the integral method used to solve for the scattering (Williams and Porter 2009), the number of modes used to establish the scattering kernel ($N_k = 100$ in our case) is distinct from the number of modes used to expand the potential ($N_v = 2$), once determined. We found that beyond the first evanescent mode ($N_v = 1$), adding them slowly deteriorate the energy conservation (Figure 1). However, we decided to include the second mode as it is susceptible to have a marginal impact on the location of the strain extremum.

We have not formally assessed it, but we believe the behaviour would be similar for a larger number of floes, given the very small spread between

the successive floes, presented in Figure 1. Additionally, we note that the domain-wide energy is very well conserved whatever the number of evanescent modes considered.

[Figure]

Figure 1: Energy conservation in a five-floe setting, with 100 kernel modes. The coloured lines, from darker to lighter hues, identify floes from left to right, where the wave forcing is incident from the left. The black line characterises energy conservation over the whole domain. Thinner lines indicate 10 individual runs, with randomly placed and spaced floes, and the thicker lines the average of these runs. With non-viscous floes, we expect the difference $1 - (|R|^2 + |T|^2)$ (vertical axis) to be 0, where $R$ and $T$ are the complex reflection and transmission coefficients associated with the propagating mode, respectively.

**2.9 Section 4**

In this section you provide a physical explanation/interpretation of the results presented in Fig. 4 but not Fig. 3. It would be good to see more discussion of the results in Fig. 3; in particular, can you explain the different trends in the variability/dispersion of floe size shown in panels (c) and (d) in Fig. 3?

For the experiments related to the strain threshold (panels (a) and (c)), we believe that the apparent translation of the histogram is due to the fact that higher strains are reached further from the floe edges (as we use free edge boundary conditions), leading to more frequent longer broken-off floes. We will investigate this behaviour further.

For the experiments related to the ice thickness (panels (b) and (d)), we expect non-linear patterns as the thickness impacts both the strain undergone by the floe and the wave transmission coefficient, which translates to the under-floe wave amplitude, hence the deflection undergone by the floe and ultimately, the strain. Short waves (4 s) are much more effectively reflected by thickening ice than longer waves (Figure 2), leading to very little breakup (apparent oscillatory behaviour between 1.5 m and 2 m for $T = 4$ s on panel (d)), which can be seen on Figure 4 (b).

We will add a discussion on these trends in the revised manuscript.

[Figure]

Figure 2: Reflection coefficient ($|R|^2$) for increasing ice thickness and various wave periods.

**2.10 Figure 3**

Why did you decide to use the median floe size to characterise the average floe size (rather than, for example, a linear-weighted mean)? An explanation in the text somewhere would be useful.

Given the skewness of the distribution of floe sizes (top panel), we decided to use the combination median–interquartile range as measures of central tendency and dispersion, rather than the combination mean–standard deviation, the data being clearly non-normal. The results presented on Figure 3 are average over realisations of a single, monochromatic case: no weighting is used. This choice is clarified in the text.

**2.11 Figure 3**

Do you have any explanation for the oscillatory behaviour in panel (d) for the two shorter wave periods when the ice thickness exceeds $1.5\,\mathrm{m}$?

We do note this apparent oscillatory behaviour. For the shortest period, we believe it to be no more than a spurious effect due to the fairly low amount of breakup, enhancing the apparent variability. We will add a comment to the revised manuscript.

**2.12 P13 L295–297**

'To estimate the effect of a developed sea on the FSD $f_L$, we take the weighted average of distributions resulting from monochromatic model runs,'. This appears to be a significant model assumption to only consider single amplitude-frequency pairs at once rather than the full wave spectrum since it ignores possible interactions between the different pairs in fracturing the sea ice. What is the justification for this model approach? There needs to be some evidence presented (e.g. test cases evaluating the model using full polychromatic forcing) to show that the error resulting from this approximation is not large enough to impact the conclusions.

The assumption behind our approach is that wave periods act independently, which is in line with our linear theory. We did consider 'full polychromatic forcing' in the sense of evaluating the strain induced by each period separately, taking the weighted average, and using it to determine whether floes should break (Mokus and Montiel 2022, conference proceedings). The two approaches yield different results. However, it is not clear which one is physically more justifiable, as the strain-averaged approach assumes steady state to be reached by waves of all periods at the same time, even though longer waves propagate faster. These models would need experimental validation for additional support.

We will insist on the assumptions and underlying limitations in the revised manuscript, and mention the alternative parametrisation (strain superposition) in the discussion.

**2.13 P13 L300**

Can you comment on the sensitivity of your results to the choice of spectrum?

We ran comparisons using alternative weighting functions, represented on Figure 3. Descriptions of these functions can be found in Table 1.

The lognormal density function has, qualitatively, a shape similar to the Pierson-Moskowitz spectrum displayed as as a function of frequency. However, we obtain similar, skewed unimodal densities with a range of symmetrical weighting functions, as displayed on Figure 4.

The one case that stands out, with a lot of secondary peaks, is n_md. It corresponds to a function giving more weights to high frequency waves (5 to 15 m), which is unrealistic in the context of our model.

We will comment on this sensitivity in the revised manuscript.

[Figure]

Figure 3: Representation of functions used as alternative weights. They are all normalised on the positive real half-line; because of the finite range of frequencies supported by our model, truncations imply some of them integrate to less than unity. Description of the legend entries can be found in Table 1.

**2.14 P14 L310**

[Figure]

Figure 4: Densities obtained when combining monochromatic model runs with different weighting functions, as represented on Figure 3.

What is the reason for drawing a single FSD $f_l$ at random rather than including all 50 realisations?

For each wave period, we obtain 50 distinct FSDs $\tilde{f}_L$. We select at random one of each before combining them, according to Eq. (23), into a FSD $f_L$. Proceeding like so allows us to observe variations between different $f_L$, as represented on Fig. 5 of the manuscript, and virtually gives us infinitely many ways to build $f_L$ ($50^{200} \approx 6 \times 10^{339}$, as we have 50 realisations of 200 periods).

Table 1: Details of several weighting methods.

| Name | Type | Effective sample size |
|------|------|----------------------|
| pm | Pierson-Moskowitz, $T_p = 5\,\mathrm{s}$ | 23451 |
| n_0 | Gaussian, $\mu = 0.2\,\mathrm{Hz}$, $\sigma = 0.05\,\mathrm{Hz}$ | 42859 |
| n_md | Gaussian, $\mu = 0.4\,\mathrm{Hz}$, $\sigma = 0.05\,\mathrm{Hz}$ | 550 |
| n_mh | Gaussian, $\mu = 0.1\,\mathrm{Hz}$, $\sigma = 0.05\,\mathrm{Hz}$ | 65347 |
| n_dd | Gaussian, $\mu = 0.2\,\mathrm{Hz}$, $\sigma = 0.1\,\mathrm{Hz}$ | 71280 |
| n_dh | Gaussian, $\mu = 0.2\,\mathrm{Hz}$, $\sigma = 0.025\,\mathrm{Hz}$ | 19158 |
| uni | uniform | 77486 |

**2.15 Section 5.3**

As it currently exists, I am not sure this section is adding much insight to the manuscript and it could be removed without detracting from the paper. All this section demonstrates is that the average floe size increases moving away from the ice edge, a behaviour several previous observational and modelling studies have identified. What might make this section more insightful would be if the results could be used to generate a mathematical description of how the emergent FSD changes with distance from the ice edge or plots to show how the parameters of the lognormal fit change with distance from the ice edge.

We added this subsection as we found the displayed features interesting. We agree it can be further extended. Most importantly from our perspective is that the behaviour stays lognormal. In the revised manuscript, we will change our non parametric density estimates to lognormal fit and add the evolution of the parameters with respect to distance from the edge.

**2.16 P16 L364–365**

'For simplicity, even though we did conduct multivariate simulations, we focus here'. Why mention this if you are not going to discuss the results? It would be beneficial to discuss some of these results—since in the results you present much of the parameter space is unexplored leaving open the potential for different behaviour elsewhere in the parameter space.

We acknowledge that some new behaviour may emerge from multivariate simulations. We will alter the manuscript to state that such simulations are outside the scope of the study.

**2.17 P17 L382–384**

'As the peak propagating wavelength is proportional to the significant wave height, this non-monotonic evolution does not support wave properties alone govern the dominant floe size,'. Can you provide a more precise explanation of why this happens? Given the simplified model treatment used, it should be possible to explain how this behaviour emerges.

We believe this is due to the period-averaging. For small waves, wave periods leading to the most breakup do not match the wave periods dominating the spectrum. We will investigate this behaviour further.

**2.18 Section 6**

It would be good to see more focus in the discussion/conclusions on what needs to be done to validate this model using observations of floe size i.e. what are the key emergent features of the FSD produced by this model that could potentially be identified in observations (not just the general lognormal shape, but how the distribution evolves with changes to key parameters such as the distance from the ice edge).

We are currently engaged in discussions with experiment-focused teams regarding that matter. We will expand this Section accordingly. We would also like to highlight the lognormal model proved to be successful at describing observation data (Montiel and Mokus 2022), even though these could not be linked to wave activity.

**2.19 P21 L446–447**

'These results aim at being a step towards the parametrisation of wave action in FSD-evolving models.' Working towards this parametrisation seems to be a key result of this study and merits more than a single line in the discussion/conclusion section. What more needs to be done to develop this parameterisation? How will this parameterisation compare to the alternative scheme developed by (Horvat and Roach 2022)?

Results can be used to, e.g., parametrise the distribution of floe sizes used in the wave fracture parametrisation implemented in CICE by (Roach et al. 2018) and labelled SP-WIFF in the aforementioned reference. The current scheme relies on an histogram ($A(r)$ in Horvat and Roach 2022) derived from a strain-based breakup scheme applied on an ice transect, i.e. a configuration with geometry assumptions similar to the model we present here. Binned floe sizes are computed off-line for a number of ice and wave conditions. A parametric distribution, whose parameters may depend on these ice and wave conditions, could be used as an alternative to this ad-hoc breakup parametrisation.

We will expand on this in the text.

**3 Technical Corrections**

**3.1 P2 L30–31**

'In particular, fragmentation caused by ocean waves makes the floes more sensitive to melt'. Maybe change 'In particular' to 'Of particular interest here' or something similar, since there exists other mechanisms of ice fragmentation that can drive the same feedback.

'Of particular interest here' conveys the intended meaning and we made the change.

**3.2  P2 L36–37**

Most studies listed fit the observed FSD to a simple power law (or combination of the two). I am not sure it is correct to describe these as Pareto distributions (see e.g. Herman, 2010).

In Herman (2010), the author called densities proportional to $x^{-1-\alpha} \exp\left(\frac{1-\alpha}{x}\right)$ truncated Pareto distributions, or GLV distributions. Without any constraint on the normalisation term, it is actually an inverse gamma distribution, whose density is $\frac{1}{\Gamma(\alpha)} x^{-(1+\alpha)} \exp\left(-\frac{1}{x}\right)$ when setting the scale to 1. The Pareto distribution probability density function is $\alpha x^{-(\alpha+1)}$, when setting the scale to 1 ($\alpha > 0$), and often abusively called power-law distribution.

**3.3  P2 L37–38**

'However, a variety of processes such as failure from wind or internal stress, lateral melting or growth, ridging, rafting or welding, are susceptible to alter the FSD.' Can you provide references for these processes having been observed to influence the FSD?

These processes are listed by Rothrock and Thorndike (1984) in their seminal paper. Many studies treating the floe size distribution reference them without further justification by observation. Some modelling studies give them a mathematical treatment as well (Horvat and Tziperman 2015). For more accuracy, we will change our statement to highlight its conjectural nature.

**3.4  P2 L48–49**

'evaluate the impact of its introduction on other quantities such as ice thickness or concentration (Roach et al., 2018)'. There are other studies you should consider referencing here e.g. Bateson et al., 2020; Boutin et al. 2021.

We will add these references.

**3.5  P3 L92**

'ensuing'. Should this be ensuring?

We will change it to 'leading to' which might better convey the intended meaning.

**3.6 P4 L96**

Reference is incorrect. Boutin et al. (2020b) should be Boutin et al. (2021).

Indeed, we referenced the Discussion paper, this will be corrected.

**3.7 P9 L243–244**

'Hence, the number of floes at most doubles, if all the floes break in a single simulation.' If my understanding is correct, single iteration would be a clearer choice here rather than single simulation.

In this subsection, we try to present the breakup scheme in a general manner. Using 'iteration' conveys the idea of repeating breakup event. It is, indeed, what we present in the following subsection; but it has not been introduced so far. This is why we settled on using 'simulation', as in 'breakup simulation'.

**3.8 P18 L394–395**

'The prevalence of smaller floes, however, tends to build up slightly.' Phrasing here is awkward.

We can replace it with 'The fraction of floes in the smallest size categories tends to increase'.

**3.9 P19 L399**

'shows'/'points out' rather than 'point out'.

Corrected.

**References**

Bateson, A. W., D. L. Feltham, D. Schröder, L. Hosekova, J. K. Ridley, and Y. Aksenov (Feb. 2020). "Impact of sea ice floe size distribution on seasonal fragmentation and melt of Arctic sea ice". In: *The Cryosphere* 14.2, pp. 403–428. DOI: 10.5194/tc-14-403-2020.

Boutin, G., T. Williams, P. Rampal, E. Olason, and C. Lique (Jan. 2021). "Wave–sea-ice interactions in a brittle rheological framework". In: *The Cryosphere* 15.1, pp. 431–457. DOI: 10.5194/tc-15-431-2021.

Herman, A. (June 2010). "Sea-ice floe-size distribution in the context of spontaneous scaling emergence in stochastic systems". In: *Physical Review E* 81.6. DOI: `10.1103/physreve.81.066123`.

Horvat, C. and E. Tziperman (Nov. 2015). "A prognostic model of the sea-ice floe size and thickness distribution". In: *The Cryosphere* 9.6, pp. 2119–2134. DOI: `10.5194/tc-9-2119-2015`.

Horvat, C. and L. A. Roach (Jan. 2022). "WIFF1.0: a hybrid machine-learning-based parameterization of wave-induced sea ice floe fracture". In: *Geoscientific Model Development* 15.2, pp. 803–814. DOI: `10.5194/gmd-15-803-2022`.

Mokus, N. and F. Montiel (Apr. 2022). "Floe size distributions in irregular sea states". In: *Proceedings of the 37th International Workshop on Water Waves and Floating Bodies.*

Montiel, F. and N. Mokus (2022). "Theoretical framework for the emergent floe size distribution: the case for log-normality". In: DOI: `10.48550/ARXIV.2205.06014`.

Roach, L. A., C. Horvat, S. M. Dean, and C. M. Bitz (June 2018). "An Emergent Sea Ice Floe Size Distribution in a Global Coupled Ocean-Sea Ice Model". In: *Journal of Geophysical Research: Oceans* 123.6, pp. 4322–4337. DOI: `10.1029/2017jc013692`.

Rothrock, D. A. and A. S. Thorndike (1984). "Measuring the sea ice floe size distribution". In: *Journal of Geophysical Research* 89.C4, p. 6477. DOI: `10.1029/jc089ic04p06477`.

Stern, H. L., A. J. Schweiger, J. Zhang, and M. Steele (Jan. 2018). "On reconciling disparate studies of the sea-ice floe size distribution". In: *Elementa: Science of the Anthropocene* 6. Ed. by J. W. Deming and T. Maksym. DOI: `10.1525/elementa.304`.

Williams, T. and R. Porter (July 2009). "The effect of submergence on the scattering by the interface between two semi-infinite sheets". In: *Journal of Fluids and Structures* 25.5, pp. 777–793. DOI: `10.1016/j.jfluidstructs.2009.02.001`.

Zhang, J., H. Stern, B. Hwang, A. Schweiger, M. Steele, M. Stark, and H. C. Graber (Jan. 2016). "Modeling the seasonal evolution of the Arctic sea ice floe size distribution". In: *Elementa: Science of the Anthropocene* 4. Ed. by J. W. Deming and T. Maksym. DOI: `10.12952/journal.elementa.000126`.

---

## Author Comment (AC4)

We thank you for your kind comments and suggestions. We address your concerns below.

**1 Methods**

**1.1 Strain parametrisation**

Using the location of the extremum strain as the location of fracture is indeed arbitrary. It comes with the simplicity of yelding an absolute answer to the breakup point question. As this location is phase-dependent, We believe that the extensive randomisation we set up mitigates the effects of this choice on the FSD.

**1.2 Relationship between fracture location, ice and wave properties**

We have not attempted to establish such a relationship. We meant to focus on the emerging distribution, rather than on demonstrating results that could be applied to single floes. Even though Having precise, deterministic results connecting ice mechanical properties and a prescribed wave forcing would be captivating, it feels less in line with moving to larger scales, which are inherently stochastically driven.

**1.3 Young's modulus**

We chose the value of 6 GPa in line with previous studies (Kohout and Meylan 2008; Williams et al. 2013). We do not attempt to evaluate its impact on our results.

**2 Monochromatic forcing**

For the right-boundary semi-infinite floe, strain is embedded in an envelope, as shown in Kohout and Meylan (e.g. 2008) and displayed in Figure 1. For finite floes, the free edge boundary condition makes the strain go to 0 on both edges. Within this envelope, strain being a superposition of propagating, attenuated and evanescent modes, oscillates with a wavelength close to the main propagating mode. It is exponentially attenuated, in the direction of propagation, in relation with the chosen viscosity; no attenuation exist for zero viscosity simulations. The amplitude of these oscillations is proportional to the wave amplitude, hence it diminishes with successive wave reflections.

[Figure]

Figure 1: Along-floe strain envelope evolution, for various ice thicknesses, $T = 8\,\mathrm{s}$, $a = 50\,\mathrm{cm}$. The first stress in excess of our reference strain threshold $\varepsilon_c = 4 \times 10^{-5}$ is located by a dot on each line.

**3  Main comment**

Displayed distributions are indeed so-called number FSDs. The same analysis can be applied to areal FSDs. If a relationship is known, or assumed, between metrics of length and floe area, going back and forth between the two is straightforward. In order to not make such an assumption, we stuck to displaying number-based results.

We present a comparison between number FSD and areal FSD in Figure 2. We obtain the second by assuming floe area to be directly proportional to the square of floe length (which holds for e.g. rectangular or elliptical floes with constant aspect ratio). It can be shown that if a random variable $X$ follows a two-parameter lognormal distribution, then powers of $X$ also follow lognormal distributions, whose parameters depend on the original parameters and the power used. If $X$ follows a three-parameter lognormal distribution, then powers of $X$ follow linear combinations of lognormal distributions; the larger the location shift, the further these combinations would be from a pure lognormal. Therefore, if the floe lengths, when considering their frequency of observation, are lognormally distributed with a small shift, we expect the

areal FSD to be close to lognormal as well.

[Figure]

Figure 2: Comparison between number (ND) and area (NA) FSDs, and overlaid lognormal fits.

**4 Forecast based on fitted parameters**

We do not attempt to fit analytical trends to the lognormal parameters. Their evolutions with respect to the model physical parameters suggests that it would be a reasonable exercise to do so and we will consider it.

**5 Discussion and conclusion**

This omission is due to a a submission timing. We do cite Dumas-Lefebvre and Dumont (2020), and we will reference Dumas-Lefebvre and Dumont (2021) in the revised manuscript.

**References**

Dumas-Lefebvre, E. and D. Dumont (2021). "Aerial observations of sea ice break-up by ship waves". In: *The Cryosphere Discussions* 2021, pp. 1–26. DOI: `10.5194/tc-2021-328`.

Dumas-Lefebvre, E. and D. Dumont (Feb. 2020). "Aerial observations of sea ice breakup by ship-induced waves". en. In: *ArcticNet Annual Scientific Meeting.* DOI: `10.13140/RG.2.2.23493.40164`.

Kohout, A. L. and M. H. Meylan (Sept. 2008). "An elastic plate model for wave attenuation and ice floe breaking in the marginal ice zone". In: *Journal of Geophysical Research* 113.C9. DOI: `10.1029/2007jc004434`.

Williams, T. D., L. G. Bennetts, V. A. Squire, D. Dumont, and L. Bertino (Nov. 2013). "Wave–ice interactions in the marginal ice zone. Part 1: Theoretical foundations". In: *Ocean Modelling* 71, pp. 81–91. DOI: `10.1016/j.ocemod.2013.05.010`.